# Causal inference from cross-sectional earth system data with geographical convergent cross mapping

Bingbo Gao[1,2,10], Jianyu Yang[1,2,10], Ziyue Chen ●[3] ✉, George Sugihara[4], Manchun Li[5], Alfred Stein[6], Mei-Po Kwan ●[7,8] & Jinfeng Wang ●[9] ✉

Causal inference in complex systems has been largely promoted by the proposal of some advanced temporal causation models. However, temporal models have serious limitations when time series data are not available or present insignificant variations, which causes a common challenge for earth system science. Meanwhile, there are few spatial causation models for fully exploring the rich spatial cross-sectional data in Earth systems. The generalized embedding theorem proves that observations can be combined together to construct the state space of the dynamic system, and if two variables are from the same dynamic system, they are causally linked. Inspired by this, here we show a Geographical Convergent Cross Mapping (GCCM) model for spatial causal inference with spatial cross-sectional data-based cross-mapping prediction in reconstructed state space. Three typical cases, where clearly existing causations cannot be measured through temporal models, demonstrate that GCCM could detect weak-moderate causations when the correlation is not significant. When the coupling between two variables is significant and strong, GCCM is advantageous in identifying the primary causation direction and better revealing the bidirectional asymmetric causation, overcoming the mirroring effect.

A sufficient and precise understanding of causal associations between the target variable and influencing variables is an important component for effectively utilizing natural resources and achieving sustainable development. Natural and social experiments are reliable approaches for proving the existence of causation between the cause and effect variable. However, in Earth system sciences, it is usually not feasible to conduct experiments to explore the causal associations between the target and influencing variables at large spatiotemporal scales[1,2]. In this case, investigation on observational data has become a commonly acceptable alternative to infer reliable causal associations. Traditional statistics is a mainstream approach to analyze observational data, yet it denied a hypothesis of causation until the 1980s[3,4]. In the past decade, causal inference has become a topic of interest in many disciplines and a diversity of approaches has been proposed, implemented and adapted for better attributing major social, ecological, and economic issues[1,5–8].

[1]College of Land Science and Technology, China Agricultural University, Beijing, China. [2]Key Laboratory of Remote Sensing of Agricultural Disasters, Ministry of Agriculture and Rural Affairs, Beijing, China. [3]Faculty of Geographical Sciences, Beijing Normal University, Beijing, China. [4]Scripps Institution of Oceanography, University of California, San Diego, La Jolla, CA, USA. [5]School of Geography and Ocean Science, Nanjing University, Nanjing, China. [6]Faculty of Geo-Information Science and Earth Observation (ITC), University of Twente, Enschede, The Netherlands. [7]Department of Geography and Resource Management, and Institute of Space and Earth Information Science, The Chinese University of Hong Kong, Hong Kong, China. [8]Department of Human Geography and Spatial Planning, Utrecht University, Utrecht, the Netherlands. [9]State Key Laboratory of Resources and Environmental Information Systems, Institute of Geographic Sciences and Nature Resources Research, Chinese Academy of Sciences, Beijing, China. [10]These authors contributed equally: Bingbo Gao, Jianyu Yang. ✉e-mail: zychen@bnu.edu.cn; wangjf@lreis.ac.cn

Structural causal modeling, which originated from Wright[9] and was developed by Pearl[10,11], and the potential outcome frameworks, which originated from Neyman and Fisher and was developed by Rubin[12,13], are the two main frameworks to identify the existence of causal relationships and estimate causal effects (the strength of the causal relationship). The former emphasizes the combination of causal graphs and statistical tools, while the latter attempts to imitate the randomized controlled trials (RCT) by adjusting the variables. Causal network learning algorithms developed from the structural causal modeling framework, such as PCMCI[1,2], can automatically build the causal networks and estimate casual effects. The potential outcome framework was proposed to embed the instrumental variables to eliminate the confounders[14] and has been widely used in social and economic science[4,15]. These two frameworks are mainly for stochastic processes. But for earth system sciences, many systems (e.g., climate system or ecosystems) contain deterministic or dynamical interactions, which are not captured well by the assumptions of these frameworks and thus challenge their applicability. For those nonlinear complex systems, causal inference models based on the theory of state space reconstruction were developed. Amongst these models[16], Convergent Cross-mapping (CCM) has been well accepted and massively employed in a diversity of disciplines[17], including epidemic diseases[18], socio-economic issues[19], and atmospheric pollution[20]. According to Takens' embedding theory[21], CCM employs time series data to reconstruct the attractor, based on which the causation can be identified and measured through the cross-mapping prediction. In addition, other models which extract causation following a similar principle were also developed, such as cross-mapping smoothness (CMS)[22]. and partial cross-mapping (PCM)[23].

However, these models can solely work with time series data, and cannot be applied to spatial cross-sectional data (the characteristics, principles and disadvantages of mainstream causation models are briefly introduced in Supplemental Section 1 and Supplementary Table 1). Spatial cross-sectional data, which often includes rich spatial information but has little information on temporal changes, is more easily available than time series data in Earth System Science. Spatial cross-sectional data records spatial processes and their interactions, and spatial order (or variation) are important indicators for understanding causal associations, which may be ignored by temporal models due to insufficient length of time series or insufficient variation across time series. Our previous research suggested that major temporal causation models, including CCM and Granger Causality, failed to extract NPP (Net Primary Productivity) -temperature (precipitation), a clear existing causation[6]. This was mainly attributed to the slight inter-annual variations of temperature (precipitation). Meanwhile, NPP-climate association was effectively identified by spatial models, thanks to notable spatial variations of temperature and precipitation. Existing models, such as "spatial difference in difference" and "spatial regression discontinuity", may in some cases measure the average causal effect of binary treatment under strong assumptions[22,23]. They are based on the stable unit treatment value assumption (SUTVA) (or ignorability), which assumes that (1) the potential outcomes of different spatial units do not interfere with each other, and (2) all treatment levels are included so that the outcome should remain unchanged if the same treatment is assigned. However, due to spatial heterogeneity and spatial spillover[7], which means that potential outcomes of one spatial unit can influence its spatial neighbors, these assumptions are easily violated in Earth System Sciences. Another major challenge lies in the identification of the direction of causal associations, which is largely disturbed by the contemporaneous dependence between spatial variables, known as the mirroring effect[24,25]. Therefore, it remains unclear whether, and more importantly, how reliable causal associations can be inferred from spatial cross-sectional data.

Given the demand for causal inference from spatial cross-sectional data in Earth System Sciences and the limitations of existing temporal and spatial causation models, here we propose a Geographical Convergent Cross Mapping (GCCM) model by adopting dynamical systems theory and generalized embedding theory[26–30]. GCCM is capable of identifying the causal associations between spatial cross-sectional variables and estimating the corresponding causal effects. By exploring nonlinear associations, GCCM outputs can be more robust than linear correlation, which may easily lead to spurious correlation, and can identify causations neglected by Linear Non-Gaussian Acyclic Model (LiNGAM)[31], a frequently used structural causal model with enhanced assumptions. Based on three typical cases, including clearly existing causation that cannot be measured through temporal causation models, we demonstrate that GCCM can detect weak-moderate causation when the correlation is not significant. On the other hand, when the correlation between two variables is significant and strong, GCCM is advantageous of identifying the primary causal direction and better revealing the unidirectional asymmetric causation, overcoming the mirroring effect. GCCM opens up a new vision to infer causal associations from spatial cross-sectional data, by bridging Earth data with the methodologies of complex system, and can be a powerful tool for revealing nonlinear complex relationships in Earth systems.

## Results

We illustrate the implementation and interpretation of GCCM with three typical cases, demonstrating the application GCCM to different types of spatial data and causal assoiactions. The first case is to infer casual associations between the soil heavy metal and two influencing factors (the density of industry pollutants and residence) in Illinois and Indiana of US. The second case is to verify the casual associations between county-level population density and environmental factors in China. The third case consider the farmland NPP-temperature (precipitation) associations in China, which are clear existing causations but fail to be identified by mainstream temporal causation models[8]. Correlation analysis, which is the most frequently used method for understanding causal associations from spatial cross-sectional data[15] and LiNGAM[32] from the structural causal modeling framework, which is commonly used to identify the direction of causations, are employed for the comparative analysis.

### Causal inference with GCCM

Causal associations are important components of inner mechanism, which can be recognized by observing and analyzing the phenomena they present[1,33,34]. The spatial distribution is an important phenomenon to extract causal associations in supplement to temporal changes[6]. The corresponding spatial cross-sectional data record the spatial processes and their interactions, and the spatial difference (order) provides a valuable reference for understanding causal associations. There have been some famous examples of inferring causation from the spatial perspective. For instance, Charles Darwin developed the theory of evolution by noticing the spatial difference of animals among Galapagos Islands[35], while Vasili Vasilievich Dokuchaev proposed the soil formation theory (Pedogenesis) according to the spatial difference of soil along the latitude[32]. Those successes are the fruit of human wisdoms applied to spatial cross-sectional data, but only with formalized mathematical methods, can the inferring framework become easily understandable and transferable for researchers from various disciplines, or for artificial intelligence (AI) to infer causal associations from big data.

The ubiquitous complex nonlinear associations make it challenging to infer causation in Earth Systems. Fortunately, theories for dynamic system provide useful tools to reveal the nonlinear and intertwined relationships[17]. Since Lorenz proposed a strategy to address the nonlinear complex problems in the state (phase)

**a. raster data**   **b. polygon data (queen contiguity)**

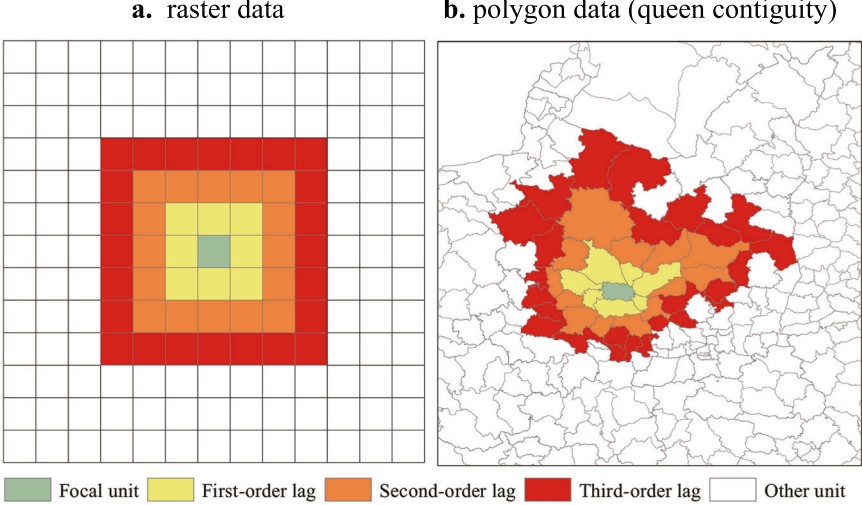

Focal unit   First-order lag   Second-order lag   Third-order lag   Other unit

**Fig. 1 | Spatial lags. a** The spatial focal unit and its spatial lags in different orders of raster data, where the first-order lags are in yellow color, the second-order in orange and the third-order lags are in red. **b** The spatial focal unit and its spatial lags in different orders of polygon data.

space[36], many prediction and causation detection methods have been developed based on Takens' theory for time-series data[21]. Takens' theory proves that, for a dynamic system $\phi$, if its trajectory converges to an attractor manifold $M$, which are consisted by a bounded and invariant set of states, then the mapping between $\phi$ and $M$ can be built and time series observations of $\phi$ can be used to construct $M$.

Earth systems are dynamic and spatial cross-sectional data can be treated as a snapshot of the dynamical system. In contrast to time series data, which are observations from a fixed spatial unit at different times, spatial cross-sectional data are observations from different spatial units at the same time. Both of them record different states of the dynamical system, only from different perspectives. The generalized embedding theorem, which is a theory on the mapping between differentiable manifold $M$ and Euclidean space $\mathbb{R}$[37], can be adopted to construct the state space with spatial cross-sectional data. Whitney proved that $M$ can be mapped into $\mathbb{R}$ by observation functions, and points in $\mathbb{R}$ can be used to construct $M$[37]. According to the theorem, for a compact $d$-dimensional manifold $M$ and a set of observation functions $\langle h_1, h_2, \ldots, h_L \rangle$, the map $\psi_{\phi,h}(x) = \langle h_1(x), h_2(x), \ldots, h_L(x) \rangle$ is an embedding of $M$ with $L = 2d + 1$. Here embedding means a one-to-one map resolving all singularities of the original manifold. The elements $h_i$ can be lags of observations from single time series observations, lags of observations from multiple time series, or multiple observation functions. The first two constructions are only special cases of the third one. By taking the measured values at one specific unit and its neighbors (named as spatial lags in spatial statistics) as a set of observation functions, $\psi_{\phi,h}(x,s) = \langle h_s(x), h_{s(1)}(x), \ldots, h_{s(L-1)}(x) \rangle$ is a embedding, where $s$ is the focal unit currently under investigation and $s(i)$ is its $i$th order of spatial lags (as shown in Fig. 1), $h_s(x)$, and $h_{s(i)}(x)$, are their observation functions respectively. Hereinafter, we will use $\psi(x,s)$ to present $\psi_{\phi,h}(x,s)$ for short. In time series data, the lag $k$ means a shift from the observation at focal period $t$ to past observation at $t{-}k\tau$. Similarly, in spatial data, the lag means a shift to spatial neighbors from the focal spatial unit. For raster data as illustrated in Fig. 1a, the first-order lags are adjacent units in eight directions in yellow color, whose first-order lags (removing those units already included) in turn constitute the second-order lags of the focal units. Similarly, for polygon data in Fig. 1b, the first-order lags are adjacent units sharing common edges or vertexes with the focal unit, and the lags of next order are the first-order lags of those adjacent units (excluding those already included). As the spatial lags in each order contain more than

one spatial units, the observation function can be set as the mean of the spatial units or other summary functions considering the spatial direction, to assure the one-to-one mapping of the original manifold $M$.

For two spatial variables $X$ and $Y$ on the same set of spatial units, organized as regular grids (raster data) or irregular polygons (vector data), their values and spatial lags can be regarded as observation functions reading values from each spatial unit. Following the generalized embedding theorem[26–30,37], their shadow manifold $Mx$ and $My$ can be constructed by assembling above-defined $\psi(x \text{ or } y, s)$. According to the dynamical systems theory[17], if $X$ and $Y$ are observed from the same dynamical system, they are governed by the same manifold. Consequently, the one-to-one map between the reconstructed $Mx$ and $My$ can be deduced because they resolve all trajectories of the same original manifold without crossings. Set one point $\psi(x,s)$ in $Mx$ as a focal state, its corresponding state $\psi(y,s)$ in $My$ can be acquired according to the mutual spatial location $s$, as well as its close states. For example, $\psi(x,s_1)$, $\psi(x,s_2)$, $\psi(x,s_3)$, and $\psi(x,s_4)$ in $Mx$ can be identified as close states of $\psi(x,s)$ according to the distance in the state space. The corresponding states of them in $My$ are $\psi(y,s_1)$, $\psi(y,s_2)$, $\psi(y,s_3)$, and $\psi(y,s_4)$ respectively based on the mutual spatial locations. Due to the mutual evolution, in $My$, $\psi(y,s_1)$, $\psi(y,s_2)$, $\psi(y,s_3)$ are also close neighbors of $\psi(y,s)$ as illustrated in Fig. 2a. Therefore, for a given $x$, the value of $y$ can be predicted according to its close neighbors identified from $Mx$. This type of prediction based on nearest mutual neighbors is defined as cross-mapping prediction, as Eq. (1).

$$\hat{Y}_s|M_x = \sum_{i=1}^{L+1}(w_{si}Y_{si}|M_x) \tag{1}$$

where $s$ represents a spatial unit at which the value of $Y$ needs to be predicted, $\hat{Y}_s$ is the prediction result, $L$ is the number of dimensions of the embedding, $si$ is the spatial unit used in the prediction, $Y_{si}$ is the observation value at $si$ and simultaneously the first component of a state in $My$, noted as $\psi(y,s_i)$. In further, $\psi(y,s_i)$ is determined by its one-to-one mapping point $\psi(x,s_i)$, which is in turn one of the $L+1$ nearest neighbors of the focal state $\psi(x,s)$ in $Mx$. $w_{si}$ is the corresponding weight defined in Eq. (2).

$$w_{si}|M_x = \frac{weight(\psi(x,s_i), \psi(x,s))}{\sum_{i=1}^{L+1} weight(\psi(x,s_i), \psi(x,s))} \tag{2}$$

**a.** The mutual neighbors of reliable prediction

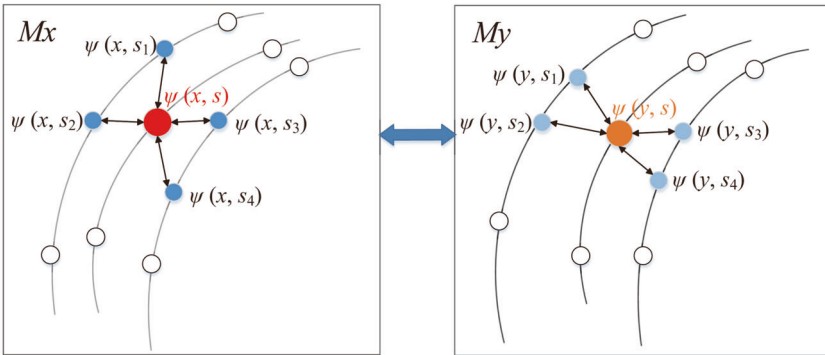

**b.** The mutual neighbors of unreliable prediction

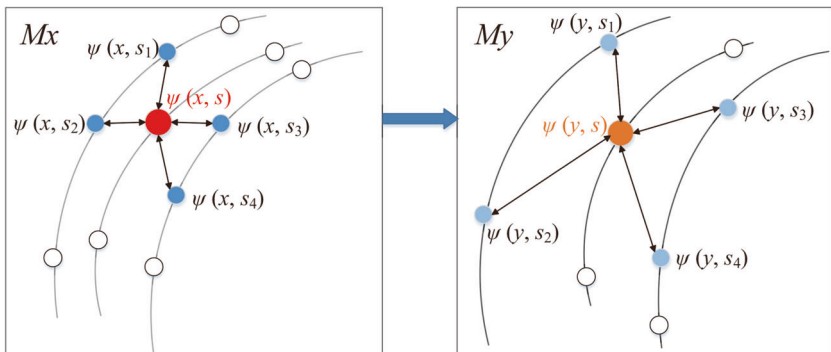

**c.** The phare space of unidirectional associations

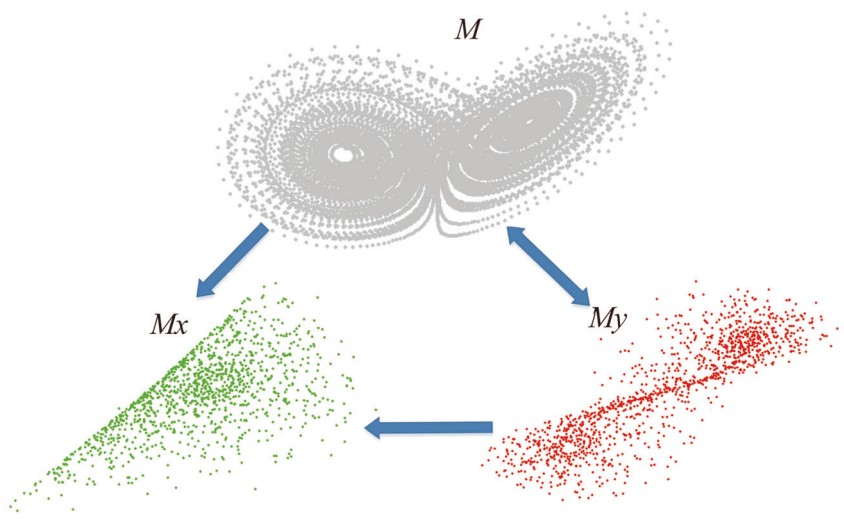

**Fig. 2 | Mutual neighbors for cross-mapping prediction. a** The mutual neighbors in the reconstructed manifold of reliable cross-mapping prediction. The orange point labeled $\psi(y,s)$ is the focal state to be predicted, and the four blue points $\psi(y,s_1)$, $\psi(y,s_2)$, $\psi(y,s_3)$, and $\psi(y,s_4)$ are nearest neighbors joining the prediction, which are found by the one-to-one mapping between $Mx$ and $My$. $\psi(x,s)$ is corresponding state of $\psi(y,s)$ in $Mx$. $\psi(x,s_1)$, $\psi(x,s_2)$, $\psi(x,s_3)$, and $\psi(x,s_4)$ are the nearest neighbors of $\psi(x,s)$ searched in $Mx$, and can be used to identify $\psi(y,s_1)$, $\psi(y,s_2)$, $\psi(y,s_3)$, and $\psi(y,s_4)$ in $My$ with mutual spatial locations. **b** The mutual neighbors of unreliable prediction. For near neighbors of focal state $\psi(x,s)$ in $Mx$, their corresponding states in $My$ are not close to the $\psi(y,s)$ in $My$. **c** The phase space of unidirectional associations. $Mx$ is a lower dimensional sub-manifold of the complete system and $My$ is a one-to-one map of $M$.

where $weight(*,*)$ is the weight function between two states in the shadow manifold, defined as Eq. (3).

$$weight\left(\psi(x,s_i),\psi(x,s)\right) = \exp\left(-\frac{dis(\psi(x,s_i),\psi(x,s))}{dis(\psi(x,s_1),\psi(x,s))}\right) \quad (3)$$

where exp is the exponential function and $dis(*,*)$ represents the distance function between two states in the shadow manifold defined

in Eq. (4).

$$dis(\psi(x,s_i),\psi(x,s)) = \frac{1}{L}\left(|h_{si}(x)-h_s(x)| + \sum_{k=1}^{L-1} abs\left[h_{si(k)}(x), h_{s(k)}(x)\right]\right) \quad (4)$$

where $|*|$ means the absolute value of a real number, and $abs[*,*]$ represents the distance function between two vectors, as the first

element $h_{si}(x)$ in $\psi(x,s_i)$ corresponds to the spatial focal units, while other elements in $\psi(x,s_i)$ respectively correspond to a vector with several spatial units. For example, $h_{s(1)}(x)$ in Fig. 1(a) has eight members in yellow color, $h_{s(1)}(x) = \langle u_{nw}, u_n, u_{ne}, u_e, u_{se}, u_s, u_{sw}, u_s \rangle$. The concrete form of $abs[*,*]$ for raster data and polygon data are specified as $abs_r$ and $abs_v$ in Eq. (5) and Eq. (6) respectively.

For raster data, as the number and location of neighbors in a certain order are fixed, $abs_r[*,*]$ can be defined as the averaged absolute difference of each spatial unit with the consideration of anisotropy. For polygon data, since the number and direction of neighbors in a fixed order of two spatial focal units differ frequently, $abs_v[*,*]$ is defined as the absolute difference of spatial lags. When the anisotropy can be neglected, Eq. (6) can also be used as the absolute value function of raster data.

$$abs_r\left[h_{si(k)}(x), h_{s(k)}(x)\right] = \frac{1}{D}\sum_{d}^{D}\left|u_{si(k,d)}(x) - u_{s(k,d)}(x)\right| \quad (5)$$

$$abs_v\left[h_{si(k)}(x), h_{s(k)}(x)\right] = \left|\frac{1}{D_1}\sum_{d}^{D_1}u_{si(k,d)}(x) - \frac{1}{D_2}\sum_{d}^{D_2}u_{s(k,d)}(x)\right| \quad (6)$$

where $u_{si(k,d)}(x)$ is the spatial unit of the $k$th-order spatial lags of $si$ in the direction $d$. $D$ is the number of spatial units (or directions) in the $k$th-order.

The skill of cross-mapping prediction is measured by the Pearson correlation coefficient between the true observations and corresponding predictions, defined in Eq. (7)

$$\rho = \frac{Cov(Y, \hat{Y})}{\sqrt{Var(Y)Var(\hat{Y})}} \quad (7)$$

where $Cov()$ represents covariance and $Var()$ represents variance.

The prediction skill $\rho$ varies by setting different sizes of libraries, which means the quantity of observations used in reconstruction of the shadow manifold. For raster data, since spatial units are regularly arranged with an equal area, the window size is used to represent the size of library. For polygon data, the number of spatial units is used to represent the size of library due to the irregular arrangement and unequal areas. Sugihara et al.[17] suggested to use the convergence of $\rho$ to infer the causal associations. For GCCM, the convergence means that $\rho$ increases with the size of libraries and is statistically significant when the library becomes largest[17,19]. By plotting a line graph of $\rho$ versus libraries' size (as Figs. 3–5), the increasing trend can be determined from it. In further, the null hypothesis for the significance test of $\rho$ is that $H_0 : \rho = 0$, and the statistics $t$ in Eq. (8) can be used with the Student t distribution to get the $p$-value. Meanwhile, the confidence interval of $\rho$ can be estimated based the statistics $z$ in Eq. (9) with the normal distribution. In following case studies, the significance level is set to 0.05 and the confidence interval is 95%.

$$t = \rho\sqrt{\frac{n-2}{1-\rho^2}} \quad (8)$$

where $n$ is the number of observations to be predicted.

$$z = \frac{1}{2}\ln\left(\frac{1+\rho}{1-\rho}\right) \quad (9)$$

If $X$ and $Y$ are from the same dynamic system, they are causally linked and can be predicted with the cross-mapping method. To assure the causal associations between $X$ and $Y$ are bidirectional or they are both the effects of a shared cause, it is a required condition that both

$\psi(x,s_i)$ and corresponding $\psi(y,s_i)$ are close neighbors of $\psi(x,s)$ and $\psi(y,s)$. In this case, with the increase of libraries, the points in the local area containing $\psi(x,s)$ (or $\psi(y,s)$) become denser and the cross-mapping prediction becomes more reliable, as demonstrated in Fig. 2a. If the causal association is unidirectional and $X$ is an external cause (noted as $X \rightarrow Y$), the original manifold contains dynamic information of both $X$ and $Y$, while $Y$ also contains information of $X$ and itself. Meanwhile, $X$ does not contain the dynamic information that solely belongs to $Y$. Consequently, $Mx$ is a lower dimensional sub-manifold of the complete system, while $My$ is a one-to-one map of $M$. It is not guaranteed that $\psi(y,s_i)$, corresponding to $\psi(x,s_i)$, is close to $\psi(y,s)$, as illustrated in Fig. 2b. So $X$ can be predicted reliably with $Y$, but not vice versa, i.e. the accuracy of the cross-mapping prediction of $X$ with $Y$ converges faster and notably stronger than the inverse effect of $Y$ with $X$, as in Figs. 3–5. If $Y$ cross-mapping predicting $X$ (i.e. noted as $Y$ xmap $X$) is higher, then we can interpreted as $X$ causes $Y$ (noted as $X \rightarrow Y$)[17]. However, if the causal influence of $X$ is strong enough that $Y$ becomes subordinated to $X$, they become synchronized (the well-known phenomenon of "synchronization"). Consequently, the capability of $X$ cross-mapping predicting $Y$ (noted as $X$ xmap $Y$) is also strong. Therefore, when the cross-mapping prediction skills are strong in both directions, it either means that the causal associations are bidirectional, or the effect variable is enslaved by the cause variable. The interpretation of GCCM outputs under different scenarios are illustrated through three following case studies.

## Extracted causations between soil pollution and multiple influencing factors

Soil heavy metal pollution exerts a strong influence on soil quality and public health[38–42]. The contents of soil heavy metals are generally stable for a long period[43] and large-scale surveys are very expensive and time-consuming[38]. Thus, the data sets of heavy metal may be updated every 10 or 20 years, making the attribution of soil pollution based on the time series data and temporal statistical models not feasible. Meanwhile, due to the wide variety of pollution sources and complex factors that influence their accumulation, the correlations between the causal factors and soil pollution are always weak and insignificant. In this study, we select four heavy metals (the soil concentrations of Cu, Cd, Mg and Pb) to present soil pollution and two influencing factors, the density of industrial pollutants and residential pollutants. The residential pollutants is usually measured by the residential density, which could be reflected by the intensity of nightlight[44,45]. Due to the lack of data concerning residential density data, the intensity of nightlight has been widely as the proxy variable of residence density[43–45]. Hereinafter in this manuscript, we use nightlight intensity to delegate the residential density. As is well known, industry and residential wastes are two major causes of soil pollution[40,46,47]. Since the ultimate challenge for verifying causation models is the lack of observable evidence, the clear existence of causation between the concentration of major heavy metals and industrial (or residential) density provides a valuable reference for assessing the reliability of causation models.

The output of causal inference using GCCM is displayed as Fig. 3. Due to limited space and similar causation outputs, here we mainly introduce the causation between Cu and the density of industrial pollutants and residents. As shown in Fig. 3a–c, the spatial distribution of soil Cu concentrations, the density of industrial pollutants and nightlight, present little similarity and shared patterns. Meanwhile, the reconstructed manifold of soil Cu concentrations, the density of industrial pollutants and nightlight, as shown in Fig. 3d–f, present a similar pattern. It demonstrates that the reconstruction process of GCCM makes some unclear spatial associations between variables clearer in the phase space. Based on the reconstructed manifold, GCCM reveals the unidirectional causation between industrial and residential density and Cu concentrations. The $\rho$ (cross-mapping prediction skill) of Cu xmap industrial density, which measures the causal

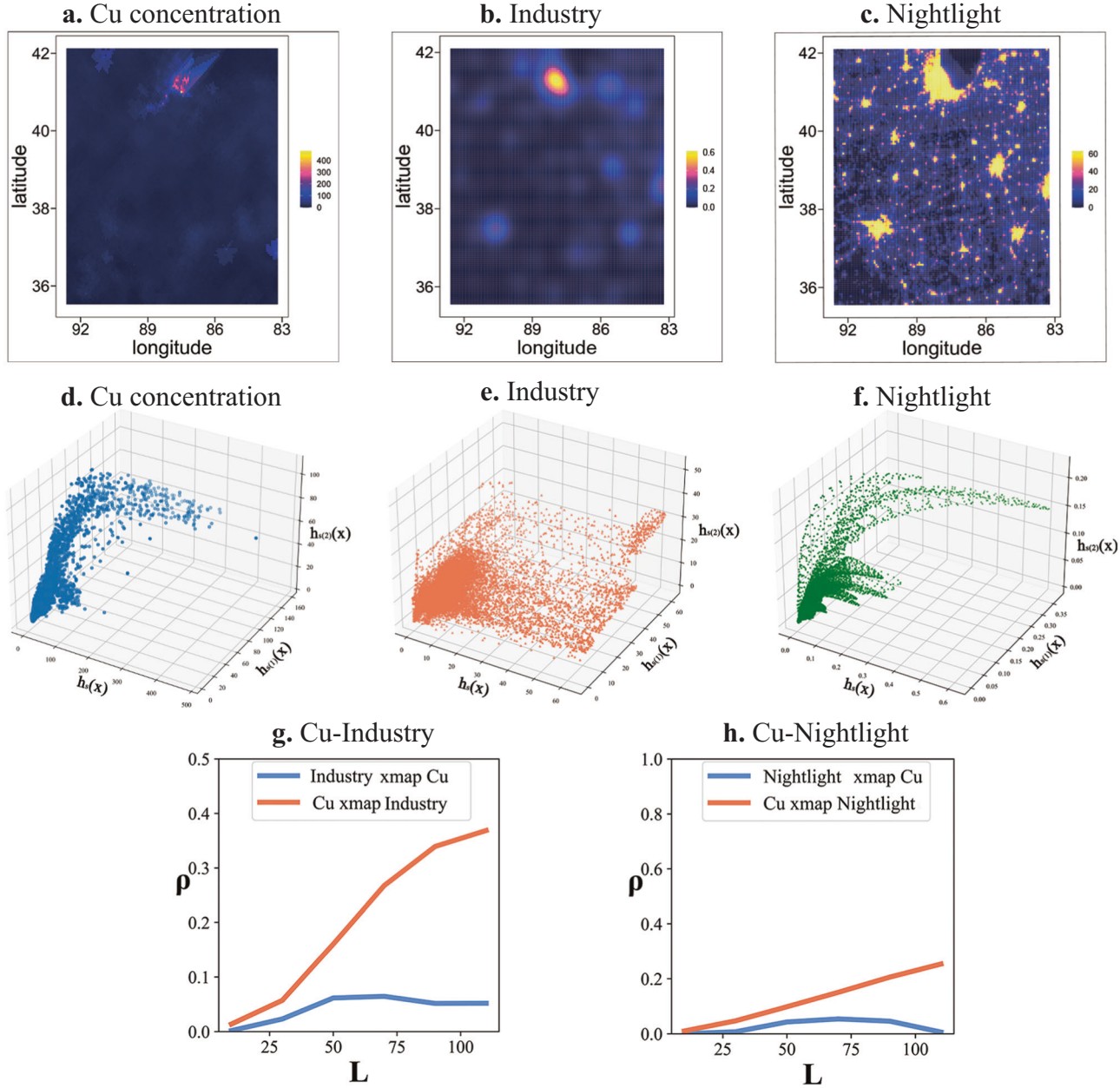

**Fig. 3 | Causal inference for soil pollution. a–c** are the maps of soil Cu concentration, the density of industrial pollutants and nightlight, **d–f** are the corresponding manifold reconstructed by spatial lags, **g** is the cross-prediction outputs between soil Cu and the density of industrial pollutant, at largest library size, $\rho$ of Cu xmap Industry is 0.37 ($p = 0.00$), and $\rho$ of Industry xmap Cu is 0.05 ($p = 0.09$), **h** is the cross-prediction outputs between soil Cu and nightlight, at largest library size, $\rho$ of Cu xmap Nightlight is 0.25 ($p = 0.00$), and $\rho$ of Nightlight xmap Cu is 0.00 ($p = 0.43$). Note: The higher Cu xmap Industry (or Nightlight) indicates a causal effect from industry (or residence) to soil pollution. As a comparison, the Pearson correlation efficient between Cu and industrial pollution density and nightlight density are −0.08 ($p = 0.43$) and −0.03 ($p = 0.08$) respectively, and the coefficients output by LiNGAM are all zero.

effect of industrial pollution density on Cu (noted as industry→Cu), in Fig. 3g, and Cu xmap nightlight, which measures the causal effect of residential pollution on Cu (residence→ Cu), in Fig. 3h are 0.37 ($p = 0.00$) and 0.25 ($p = 0.00$). Meanwhile, the $\rho$ of industrial pollution density xmap Cu (Cu→ industry) and nightlight xmap Cu (Cu→ residence) are weak and insignificant, as 0.05 ($p = 0.09$) and 0.00 ($p = 0.43$) respectively. The larger and significant $\rho$ of Cu xmap industrial density (industry→ Cu) and Cu xmap nightlight (residence → Cu) suggest that industrial and residential pollution exert a strong influence on Cu concentrations. Conversely, the lower and insignificant $\rho$ of industrial pollution density xmap Cu (Cu→ industry) and nightlight xmap Cu (Cu→ residence) suggest that Cu concentrations are not the cause of the density of industrial and residential pollutants.

Due to the lack of similarity in Fig. 3(a), (b) and (c), the commonly employed Pearson correlation analysis fail to extract significant correlations between Cu concentrations and two influencing factors, with correlation efficient as −0.08 ($p = 0.43$) and −0.03 ($p = 0.08$) for industrial pollution density and nightlight density respectively. The linear coefficients matrix (Bpruned) outputted by LiNGAM is a zero matrix, indicating that none causal associations were identified between Cu concentrations and the two influence factors. The detailed coefficients and uncertainties of GCCM, correlation analysis and LiNGAM are shown in Supplemental Table S2. The causation inference outputs of other heavy metals (Cd, Mg and Pb) are similar to Cu and can also be found in Supplemental Table S2 and Fig. S2.

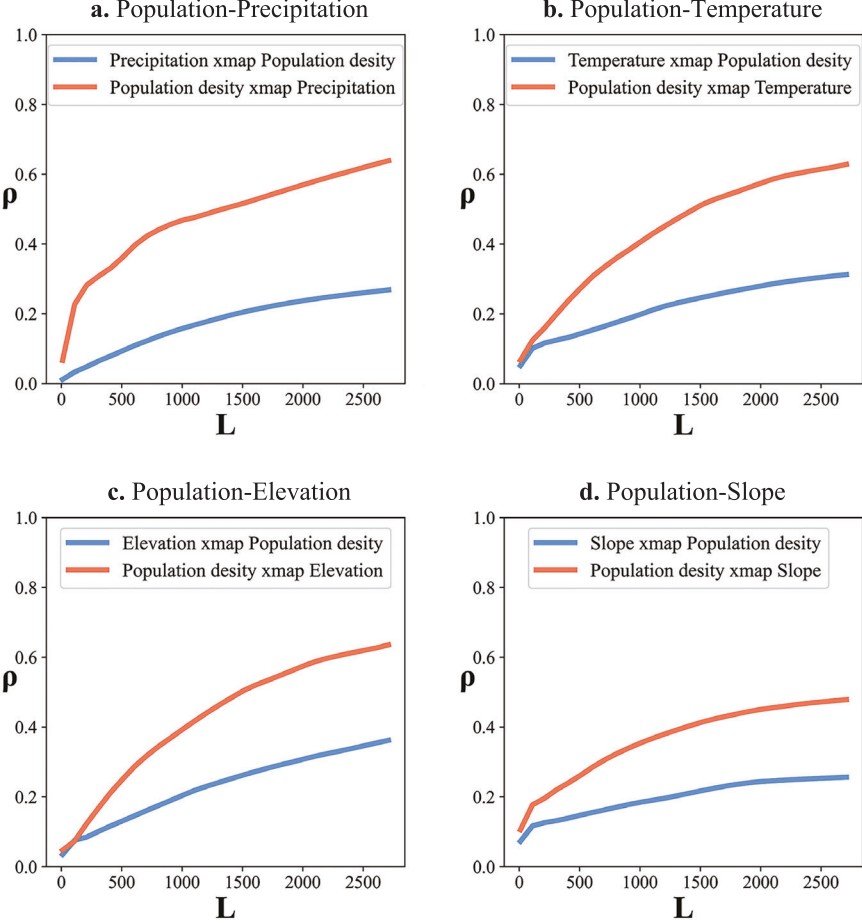

**Fig. 4 | Causal inference for population density. a–d** are the cross-mapping prediction outputs between population density and county-level Temperature, Precipitation, Elevation and Slope in China, at largest library size, $\rho$ of Population density xmap Precipitation, Temperature, Elevation and Slope are 0.64 ($p = 0.00$), 0.63($p = 0.00$), 0.64($p = 0.00$) and 0.48 ($p = 0.00$); $\rho$ of Precipitation, Temperature, Elevation and Slope xmap Population density are 0.27($p = 0.00$), 0.31 ($p = 0.00$), 0.36 ($p = 0.00$) and 0.26 ($p = 0.00$). Note: The higher Population density xmap

Precipitation (Temperature, Elevation, or Slope) indicates a causal effect from precipitation (temperature, elevation, or slope) to population density(leading casual direction). As a comparison, the Pearson correlation coefficients between population densities and precipitation, temperature, elevation and slope are 0.06 ($p = 0.00$), 0.15 ($p = 0.00$), −0.18 ($p = 0.00$) and −0.23 ($p = 0.00$) respectively, and the coefficients of LiNGAM between population density and precipitation, temperature, elevation and slope are 0.00, 0.00, −1.08 and −364.03.

This case study indicates that GCCM performs effectively to reveal the unidirectional associations between variables with weak-moderate coupling.

### Extracted causations between population density and multiple influencing factors

In this case, we examine the county-level population density organized as polygon data, to evaluate the reliability of GCCM. Different from soil pollution, we select four environmental factors closely correlated with population density, including precipitation, temperature, elevation and slope, as shown in Supplemental Fig. S3. These factors are all significantly correlated with population density (Supplemental Table S3). It is worth mentioning that similar to CCM[17], the running of GCCM also requires a non-linear relationship between variables. To address this issue, we conduct a linearity-removal pre-processing for these variables before the running of GCCM. The detailed methodology for linearity-removal can be found in the Supplemental Section S3. The causation between population density and environmental variables revealed by GCCM are displayed in Fig. 4. The $\rho$ of population density xmap precipitation, temperature, elevation and slope (precipitation→ population density, temperature→ population density, elevation→ population density and slope→ population density respectively) are large and significant, as 0.64 ($p = 0.00$), 0.63 ($p = 0.00$),

0.64($p = 0.00$) and 0.48 ($p = 0.00$). As a comparison, the $\rho$ of precipitation, temperature, elevation and slope xmap population density (population density→ precipitation, population density→ temperature, population density→elevation and population density → slope) are also significant yet much smaller, as 0.27($p = 0.00$), 0.31 ($p = 0.00$), 0.36 ($p = 0.00$) and 0.26 ($p = 0.00$). It is worth mentioning that the much smaller $\rho$ of environmental factors xmap population density (population density→ environmental factors) do not necessarily prove the existence of a reverse-direction causation. As explained by Suighara et al. [17], when there is a strong causation from $X$ to $Y$ ($X{\rightarrow}Y$), then even if there is no feedback causal influence from $Y$ to $X$, there can also be a smaller $\rho$ of $X$ xmap $Y$ ($Y{\rightarrow}X$), resulting from the subordinating effect under the strong-association scenario. Therefore, the smaller $\rho$ of $X$ xmap $Y$ can either indicate a weaker causal influence of $Y$ on $X$ (feedback causation), or the spatial variation of $Y$ can have a small reflection on the spatial variation of $X$ (reflection, not causation). The much smaller $\rho$ of environmental factors xmap population density (population density→ environmental factors) did not necessarily prove the existence of a reverse-direction causation. When there is a strong causation from $X$ to $Y$ ($X{\rightarrow}Y$), then even if there is no feedback causal influence from $Y$ to $X$, there can also be a smaller $\rho$ of $X$ xmap $Y$ ($Y{\rightarrow}X$), resulting from the subordinating effect[17]. Therefore, the smaller $\rho$ of $X$ xmap $Y$ can either indicate a weaker causal influence of $Y$ on $X$

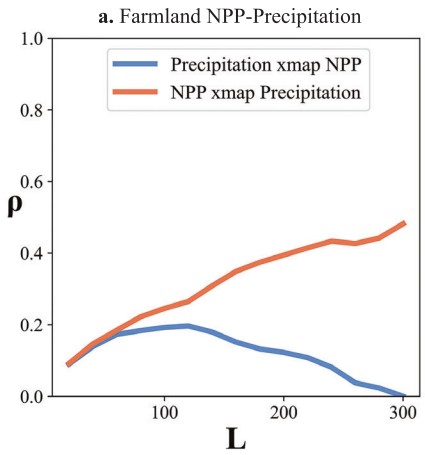
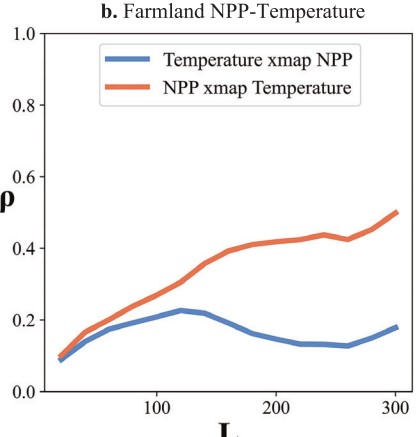

**Fig. 5 | Casual inference for farmland NPP. a** cross-mapping prediction between farmland NPP and precipitation, at largest library size, ρ of NPP xmap Precipitation is 0.48 ($p = 0.00$), and ρ of Precipitation xmap NPP is 0.00 ($p = 0.50$) **b** cross-mapping prediction between farmland NPP and temperature, at largest library size, ρ of NPP xmap Temperature is 0.50 ($p = 0.00$), and ρ of Temperature xmap NPP is 0.17 ($p = 0.00$). Note: The higher NPP xmap Precipitation (or Temperature) indicates a causal effect from precipitation (or temperature) to NPP (leading casual direction). As a comparison, the Pearson correlation coefficient between NPP and temperature and precipitation are 0.76 ($p = 0.00$) and 0.66 ($p = 0.00$), and the coefficient of LiNGAM between farmland NPP and precipitation and temperature are all zero.

(feedback causation), or the spatial variation of *Y* can have a small reflection on the spatial variation of *X* (reflection, not causation). For this research, since population density can influence the temperature through intensive human activities[43], and the precipitation by releasing particulate matters[44], which are important nuclei for rain, the small ρ of precipitation xmap population density and temperature xmap population density can be regarded as a weaker feedback causation. On the other hand, population density cannot directly alter county-level elevation. Instead, population density can partially reflect the county-level elevation. For instance, when noticing a city with a population density notably lower than other cities in China, we can presume that the elevation of this city is very likely large and located in plateau regions. So the small ρ of elevation xmap population density can be regarded as a weak reflection. Although GCCM itself cannot distinguish the feedback causation or reflection in strong coupling cases, it can identify the leading direction of the casual associations.

As a comparison, the linear correlation coefficients between population densities and precipitation, temperature, elevation, and slope are 0.06 ($p = 0.00$), 0.15 ($p = 0.00$), −0.18 ($p = 0.00$) and −0.23 ($p = 0.00$) respectively. Although the linear correlations are also significant, the coefficients of GCCM are much higher and effectively reflect the asymmetric association between two variables. As a comparison, the fixed value of the correlation coefficient cannot help understand the asymmetric direction and strength of causation between two variables. The linear coefficients of LiNGAM between precipitation, temperature, elevation and slope, and population density are 0.00, 0.00, −1.08, and −364.03. It demonstrates that LiNGAM, with linear relationship and independent non-Gaussian distributed noise assumption, fails to identify the causal associations between population density and precipitation, and temperature. The detailed results of three methods can be in found in Supplemental Table S3.

**Extracted causations between farmland NPP and climate factors**
In addition to the above cases which illustrate the performance of GCCM when processing raster data with weak correlations and polygon data with strong correlations respectively, we also check whether GCCM could effectively reveal the well-known NPP-Climate causation, a case explored in our previous study[6]. As is widely acknowledged, precipitation and temperature are the main driving forces of farmland NPP[48,49]. However, Gao et al.[6] proved that mainstream temporal causation models such as CCM and Granger Causality Test (GCT) failed to

infer causal associations between farmland NPP and precipitation (or temperature) due to insignificant temporal variations of these variables from 2000 to 2015. In this case, we employ the same data sets, including the average farmland NPP, precipitation, and temperature across China, to infer NPP-Climate associations using GCCM. Considering the strong linear correlation between NPP and precipitation (temperature), we conducted the same linearity-removal pre-processing (as in Supplemental Section S4) before the running of GCCM. The causal inference outputs from GCCM are displayed in Fig. 5, and detailed outputs of correlation analysis and LiNGAM can be found in Supplemental Table S4.

For GCCM, the prediction skill ρ of farmland NPP xmap precipitation (precipitation→ farmland NPP) and temperature (temperature→ farmland NPP) are 0.48 ($p = 0.00$) and 0.50 ($p = 0.00$). Conversely, the ρ of precipitation and temperature xmap farmland NPP (farmland NPP→ precipitation and farmland NPP→ temperature) are 0.00 ($p = 0.50$) and 0.17 ($p = 0.00$) respectively. These outputs prove a strong casual influence of precipitation and temperature on farmland NPP, which is consistent with the well-known facts, yet cannot be extracted by major temporal causation models. The zero ρ of precipitation xmap farmland NPP indicate that farmland NPP is not a cause of precipitation. And the much-smaller ρ of temperature xmap farmland NPP majorly result from the above-introduced enslaved effect from the strong causal influence of temperature on farmland NPP. In other words, farmland NPP can partially reflect temperature.

Based on the same data set, the Pearson correlation coefficient between NPP and temperature and precipitation are 0.76 ($p = 0.00$) and 0.66 ($p = 0.00$), both significant. However, this coefficient cannot help understand the asymmetric direction and strength of causation between two variables and can be largely biased due to other confounding factors[17]. As a comparison, LiNGAM fail to identify the existence of causal associations between farmland NPP and precipitation (temperature), with all coefficients as 0.00. The incapacity of LiNGAM in extracting NPP-climate causation, is mainly caused by the fact that its strong assumption on the linear relationship and noise distribution is easily violated in Earth System Science. The detailed outputs of three methods can be in found in Supplemental Table S4.

Case 2 and Case 3 suggest when variables are strongly and significantly correlated, GCCM can also extract reliable causation between them. The leading cause-effect relationship between two variables (the notably larger ρ) can be well identified and measured

through GCCM based on spatial cross-sectional data. Meanwhile, the feedback causal influence or the reflection capability from the target variable to influencing variables can as well be extracted as a smaller $\rho$, forming the unidirectional or bidirectional asymmetric associations between two variables. Since the correlation between two variables can be largely biased by other confounding variables[17], the correlation coefficient can be notably underestimated or overestimated, especially in highly complex ecosystems[17]. In this case, the $\rho$ value generated by GCCM, which is drawn from the nonlinear associations and thus more robust, can effectively reduce the influencing of other variables and provide a more reliable and complete reference for better understanding and comparing the strength of causation between the target variable and multiple influencing variables.

## Discussion

Spatial cross-sectional data is another main type of observational data of Earth systems in parallel to time series data. It contains abundant spatial information for revealing the asymmetric and multi-directional interactions between a wide range of ecological processes and massive socio-economic-environmental influencing factors. However, how to properly infer causal associations from spatial cross-sectional data remains highly challenging. Different from temporal causal models, which consider time sequence (i.e. cause precedes effect) as the fundamental and easily understandable evidence of cause-effect processes, it is more difficult for spatial models to properly link spatial variations of the target and influencing factors to their potential causations. For complex ecosystems, where there are strong, multi-direction interactions between a large number of variables, existing spatial models cannot reveal the asymmetric bidirectional causation between two variables. Against this background, GCCM is designed for spatial cross-sectional data to infer causality from spatial variations. Supported by dynamical systems theory and generalized embedding theory, GCCM suggests that similar to the time lags in time series, we could reconstruct the manifold through spatial lags and interpret the causation based on the cross-mapping predictions in the phase space. In this case, the principle of identification and measurement of causation from a spatial perspective can be clearly illustrated in the GCCM framework. Further, GCCM can reliably detect the direction and strength of causal associations in weak-to-moderate coupling systems, which may be ignored by correlation models. For strongly coupled systems, GCCM can effectively overcome the mirroring effect and thus identify the direction and the asymmetric strength of the leading causation. Facing complex systems with multiple interacting factors, similar to CCM, GCCM is advantageous in handling the non-separability issue and the nonlinear relationships. Due to the widespread associations existing between interacting elements within the Earth system, the effect variable is unlikely to be fully separated from the cause variables, and their relationships are often nonlinear. Since the commonly employed correlation model can identify neither weak nor nonlinear coupling relationships and is largely biased when dealing with strong coupling relationships, GCCM can notably improve causal inference in complex Earth systems. In addition, the associations extracted through the nonlinear relationships are more robust than linear correlations. This is because in practice, spurious linear concomitant covariations are very common and cannot be ruled out by the linear correlation. As a comparison, the probability of spurious concomitant nonlinear covariation is much lower and the associations detected by GCCM are more likely to be genuine couplings. Furthermore, GCCM is almost a parameter-free method, without the need to train many parameters, and is applicable to both polygon (vector) data and raster data, two major types of spatial cross-sectional data sources. Therefore, it could be easily employed by scholars from multiple backgrounds.

Despite its wide suitability and high reliability, some limitations remain for GCCM. The underlying assumption of GCCM is that the spatial cross-sectional variables are from dynamic systems. So it requires that information on cause variables is printed in the effect variable. If variables are from purely stochastic and linear systems and information on the cause variable is independently unique to the effect variable, GCCM becomes inapplicable. Previous studies[50–53] pointed out that causation models based on state space reconstruction were sensitive to periodic time series and high levels of noise. The former is not an issue for GCCM, as periodic spatial phenomenon are very rare in practice. On the other hand, high levels of noise in the data sources may notably affect GCCM. To evaluate GCCM when processing data with noise, we add different levels of noise to the heavy metal data set (the first case) and repeatedly run the GCCM based on the synthetic data. GCCM outputs based on the varying synthetic Cu (with added noise) are displayed in Fig. 6, and outputs of other heavy metals are presented in Supplemental Section 5. By increasing the added random noise from 10% to 30%, 60%, and finally 90% of the original observed value, the $\rho$ of Cu xmap industry pollution density and nightlight density (industry→Cu and residence→Cu) decrease gradually. Similar to other causation models, the existence of random noise in original data sources weakens the inference skill of GCCM. However, only until the random noise reaches 90% (signal-to-noise ratio approaches 1), GCCM fails to extract some causations between residence and Cu (and other heavy metals), as shown in Fig. 6 and Supplemental Section 5. To better implement GCCM, the pre-selection of data sources with limited noise and the pre-removal of noise are recommended. At the same time, some key issues for implementing and interpreting GCCM should be again emphasized here. Firstly, GCCM, as well as the classic CCM, is specially developed to infer nonlinear and intertwined casual associations from spatial perspective. Therefore, when the target and influencing factor are significantly correlated, a linear-removal processing is required before the running of GCCM. Secondly, due to the existence of enslaved association, GCCM may calculate a reverse-direction and smaller $\rho$, even if no actual feedback effect exist, and researchers should not easily explain it as so. Instead, sufficient prior-knowledge or additional investigation is required to interpret whether it is a feedback causation or it is just a reflection of the main-direction causation. Thirdly, the proxy variable "nightlight", which has been frequently considered as a representation of "residence density", in the first case is used to test the performance of GCCM to identify a known causal association. However, for identifying unknown causal association, inappropriate proxy variable may cause incorrect result, and thus some major criteria[54,55] for properly selecting proxy variables should be strictly followed.

A causal understanding of the spatiotemporal processes and mechanism and Earth System, such as climate change, environmental pollution, cultivated land degradation, healthcare, and economic problems, is crucial for human being to face challenges of survival and sustainable development. As experiments on Earth System at large scales are not feasible, causal inference based on observations has become the mainstream way to answer those needs. In recent years, although advanced causal inference has been given increasing emphasis globally, it is well accepted that existing models cannot fully establish a framework, which effectively considers the massive uncertain interactions in complex systems. Our recent research[56] revealed that some advanced models, which performed effectively in simple systems, have considerable limitations when applied to complex atmospheric environments. Meanwhile, spatial variations and temporal changes are two important perspective to infer causal associations from observations[6]. When the time series observations are absent or present insignificant changes, causal inference from spatial perspective, for which the GCCM is specially designed, can be an important complement to temporal method.

With the accumulation of spatiotemporal observations, the big data provide rich information to infer causations. To reveal the casual associations of earth system from the big data, proper integration or

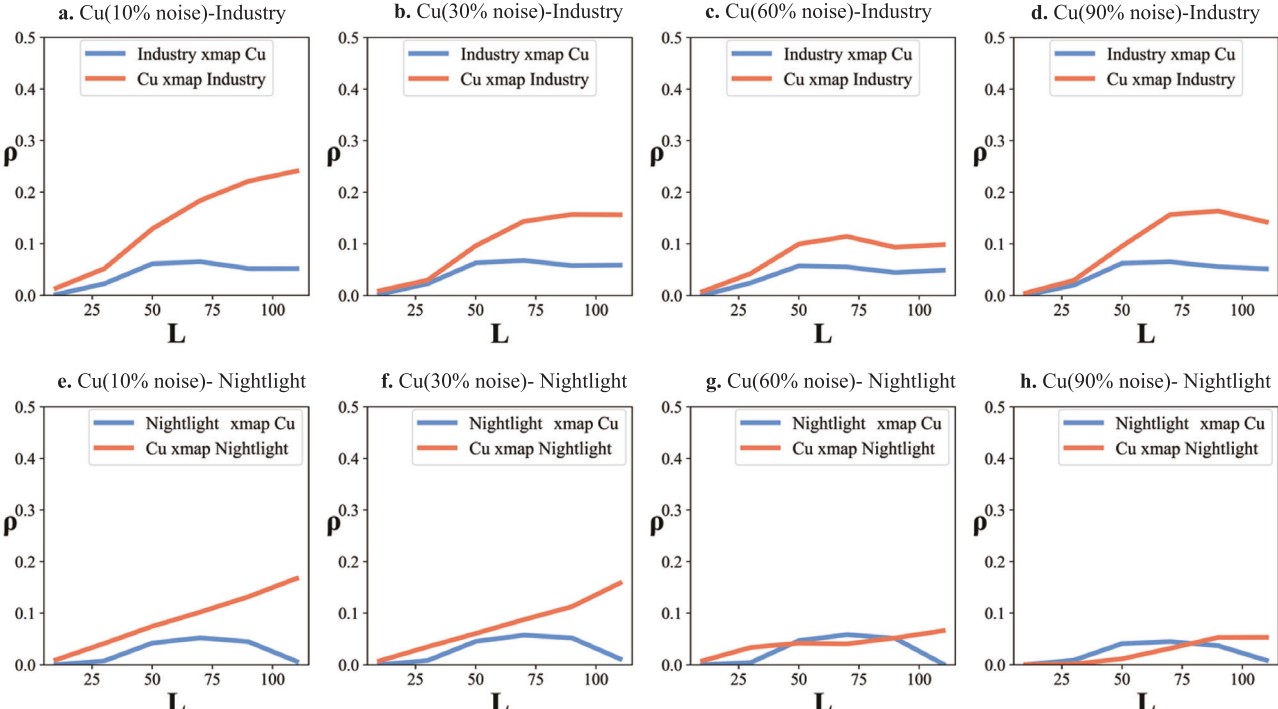

**Fig. 6 | Causal inference for synthetic Cu data with randomly added noise.** **a**–**d** are the cross-prediction results between the industrial pollution density and synthetic Cu data with 10% to 30%, 60%, and 90% randomly noise added; **e**–**h** are the cross-prediction results between nightlight and synthetic Cu with 10% to 30%, 60%, and 90% randomly noise added. Note: the higher $\rho$ of Cu xmap Industry (or Nightlight) indicates a causal effect from industry (or residence) to soil pollution.

comprehensive use of different models is recommended to handle the large number of variables and complex direct and indirect associations. In addition to its major advantages when applied alone, GCCM is a data-based causal inference method and has the potential to be directly embedded into structural causal modeling methods. For now, there are some causal network learning models (e.g. PCMCI) that can retain the correct causal graphs by leveraging the law that the cause precedes the effect in time[2]. However, for spatial cross-sectional data where information on temporal precedence is lacking, the causal direction cannot be identified even with such enhanced models as the LiNGAM. Filling this, GCCM can be effectively integrated with causal graphs for a better understanding of the complex causal relationships among variables. When we have a prior causal graph, GCCM provides a reference for further adjusting the pathways and better estimating the causal strength. When prior knowledge to build a causal graph is not available or a large number of variables have to be investigated, GCCM can be employed to improve the causal-network learning models and support the identification of causal directions and the removal of spurious Markov equivalent graphs.

In general, GCCM is an important extension of classic CCM to causal inference from a spatial perspective. CCM and GCCM can be combined to constitute a set of solutions for non-separability systems, as well as causation models for stochastic systems (e.g. Granger Causation test). In the future, scholars may further explore the possibility of embedding both CCM and GCCM into structural causal modeling methods for a comprehensive causal inference from spatiotemporal big data, leading to advanced tools for revealing the complicated human-environment interaction in the earth system.

## Methods
### Data sources
We employ three typical cases to illustrate the effects, reliability, and interpretation of GCCM in the real world. The first is the causation between soil pollution and industrial pollution and residential density

based on raster data. The second is the causation between population density and a series of environmental factors based on vector data. The third is the causation between farmland NPP and climate factors. The soil pollution data are obtained from the National Geochemical Survey database of the U.S. covering Illinois and Indiana (https://mrdata.usgs.gov/geochem/)[57]. The population density data of China are obtained from the National Bureau of Statistics of China, and the climate and topographical data are obtained from published data set[58]. The NPP data are MOD17A3V055 (http://files.ntsg.umt.edu/data/NTSG_Products/MOD17) masked with farmland of China, which are obtained from the China Multi-period Land Use Land Cover Remote Sensing Monitoring Data Set(CNLUCC)[59]. The dominant linear trend is removed from the population density and farmland NPP with the Ordinary Least Squares regression method[60,61]. The detailed setting and complete result of the experiment are presented in the supplementary documents.

GCCM is implemented with R language, with one implementation for raster data and another for polygon data. All the cases are run in R 4.2.0, the *stats* and *pcalg* package are adopted to conduct the linear correlation and LiNGAM method respectively.

### Brief steps for conducting GCCM
GCCM aims to identify and measure the causation from spatial cross-sectional data. It reconstructs the manifold in state space with spatial observations and their spatial lags, and infer causation according to the cross-mapping prediction in the state space. With the major principles and parameters of GCCM already introduced in details, the implementation of GCCM is realized in the following major steps:

1. Construct the embeddings: Firstly, set the dimension of the embeddings as $L$; Then go through each spatial unit (pixel or polygon) $s$ and find its spatial lags of different orders (as Fig. 1a, b), each of which is marked as $s(1), s(2), \ldots, s(L-1)$, and contains a set of neighboring units; Following this, get values of $X$ and $Y$ at the corresponding units and organize them as vectors,

$\langle h_s(x), h_{s(1)}(x), \ldots, h_{s(L-1)}(x) \rangle$ and $\langle h_s(y), h_{s(1)}(y), \ldots, h_{s(L-1)}(y) \rangle$; Finally, assemble those vectors together as an matrix according to their spatial orders.

2. Predict $Y$ based on $X$: Set a sequence of library sizes (i.e. the widows size for raster data and the number of spatial units for polygon data); For each library size, predict $Y$ as Eq. (1) by searching near points in the state space and confined by the library size; Calculate prediction skill $\rho$ as Eq. (7), and finally get a series of $\rho$ of $Y$ with different library sizes.

3. Predict $X$ based on $Y$: predict $X$ using the similar approach in the above step and get a series of $\rho$ of $X$ with different library sizes.

4. Extract and interpret causations: Plot the prediction outputs from above two step as line graph using $\rho$ on the vertical axis and library sizes on the lateral axis (as demonstrated in Figs. 3–6); Identify the existence, direction and strength of bidirectional (or unidirectional) causation based on the convergence, significance and confidence interval of $\rho$ with the largest library size. It should be noted, which may be misunderstood, that the line of $Y$ xmap $X$ is the basis to determine whether $X$ is a cause of $Y$ (noted as $X \to Y$)[17]. If the line of $Y$ xmap $X$ is much higher with the increase of the library size, then $X$ causes $Y$ (noted $X \to Y$) is the leading direction in a bidirectional causation, or the sole direction in a unidirectional causation.

The dimension parameter $L$ can be adjusted automatically to find the largest $\rho$. The library sizes can be commonly set as Arithmetic Sequence to reveal the trend of prediction skill.

## Data availability
The data used in this study are available at https://doi.org/10.6084/m9.figshare.21782201.

## Code availability
The codes for GCCM that we developed in this study are publicly available at https://github.com/Bingbo-Gao/GCCM.

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

## Acknowledgements
B.G. is supported by the National Key Research and Development Program of China (No. 2021YFE0102300) and the National Natural Science Foundation of China (No. 42271428), Z.C. is supported by the National Natural Science Foundation of China (No. 42171399).

## Author contributions
B.G., Z.C., G.S., and J.W. conceived of the study. B.G. and J.Y. curated the data. Z.C., B.G., and J.Y. performed the analysis. Z.C. and B.G. produced the figures. Z.C. and B.G. wrote the first draft of the manuscript. Z.C., B.G., M.L., A.S., M.K., and J.W. reviewed and edited the manuscript.

## Competing interests
The authors declare no competing interests.
