## [Peer Review File · Nature Communications]

Causal Inference from cross-sectional Earth System data with Geographical Convergent Cross MappingReviewers' comments:

Reviewer #1 (Remarks to the Author):

Summary:

In the article under review, the authors adapt convergent cross mapping (CCM), a dynamical systems-based method for inferring cause-effect relationships between time series, to spatial data. As the authors correctly point out in the introduction, although CCM has been applied in several different fields of research (e.g. the Earth sciences), it can often be difficult to obtain time series data of sufficient length, for instance if measurements only started recently compared to the time scale one is interested in. Spatial cross-sectional data on the other hand might be available more abundantly or at least can be obtained more easily. Therefore, the method presented in this work, GCCM, can be a valuable and potentially impactful contribution to the literature on causal inference for dynamical systems.

However, while the idea behind GCCM is original and the real data examples to which the method is applied are interesting, I also have serious criticism concerning the interpretation of the method's result, the discussion of the method's applicability and the presentation of the material.

Regarding the former, as I explain in more detail below, I believe that the results of the first real-data example should be interpreted more carefully. In addition, to be of real value to domain experts, any causality method should be accompanied by a thorough discussion of its underlying assumptions, its limitations, and its failure modes. Unfortunately, such a discussion is mostly lacking here. For example, CCM-based methods have been found to be vulnerable to dynamic noise, see e.g. (Bartsev et al., 2021; Cui, Moore, 2021; Krakovska et al, 2018; Cobey, Baskerville, 2016), and this is likely to be a weakness of GCCM as well. Moreover, the criteria that the method invokes (convergence and $\rho > 0$) are somewhat vague (for instance, when is the method considered to converge exactly?) and it does not come with a measure of uncertainty that conveys to a user how trustworthy the output is. Lastly, the article contains many typos and other minor issues, and the methodology section is not particularly easy to follow.

In conclusion, I unfortunately must vote to **reject** the publication of this work. Due to the novelty of the idea, a resubmission might be justified if the issues above are addressed properly.

* Bartsev, S., et al. "Imperfection of the convergent cross-mapping method." IOP Conference Series: Materials Science and Engineering. Vol. 1047. No. 1. IOP Publishing, 2021.

* Cui, Lipeng, and Jack Murdoch Moore. "Causal network reconstruction from nonlinear time series: A comparative study." International Journal of Modern Physics C 32.04 (2021): 2150049.

* Krakovska, Anna, et al. "Comparison of six methods for the detection of causality in a bivariate time series." Physical Review E 97.4 (2018): 042207.

* Cobey S, Baskerville EB (2016) Limits to Causal Inference with State-Space Reconstruction for Infectious Disease. PLoS ONE 11(12): e0169050.

Detailed comments to the authors:

- As a major motivating example, you mention the causal link between Net Primary Productivity and precipitation which can't be identified by time series-based causality methods. Yet you don't apply your method to this example and only refer to other spatial methods in the introduction. Is GCCM able to find this causal link?
- At this point, the baseline to which you compare the performance of GCCM is correlation analysis. It would be good to add another baseline method to your analysis that aims to infer causal links. This could be a naïve method, e.g. treating the spatial observations as i.i.d. observations of one random

variable and then using a bivariate causality method such as LiNGaM. I expect GCCM to clearly outperform this baseline, but it would be nice to have an empirical demonstration.

- The weaknesses and limitations of GCCM should be discussed more extensively. How vulnerable is the method to noisy data? Did you encounter real world or simulated data examples where GCCM could not identify the correct causal direction? It would be helpful if an example that is a failure mode of the method would be included to illustrate the methods limitations more clearly.
- The methods section 'The establishment of GCCM' feels a bit rushed. It would improve readability if you took more time to explain your notations and disentangle some of the sentences such as in lines 397-400. Computing different quantities explicitly for a toy example would also help.

Comments on real data examples:

I like the choice of the two real data examples. However, there are additional aspects that should be discussed.

- In the first example, the role of proxies should be discussed more clearly. What I mean is that you argue that there is a unidirectional causal link from nightlights to soil pollution whereas clearly, the fact that there is light at night does not pollute the soil. Therefore, in the interventional meaning of causation, the variable nightlights is not a cause of soil pollution. Rather, nightlights are a good proxy for an actual underlying cause, the number of residents in the area. In a way, you don't detect a causal link but a confounded connection between soil pollution and nightlights, the confounder being the number of residents. However, the variable nightlights is so informative about the confounder (i.e. such a close proxy) that it inherits the causal asymmetry between residents and soil pollution. This is a reason why the asymmetry that CCM-based methods such as yours detect should be interpreted more carefully.
- In the second example, there seems a mismatch between the labels of the plot and the causal direction that they identify (the labels should be switched). Am I right in assuming that this is not a conceptual issue but rather a mix up when that happened when generating the labels of the causal directions in the plot?
- Is there a way to assess how significant the results of your method are and how much confidence one should attribute to them? I agree that the plots in your examples look convincing but those were relatively clear-cut cases from the start. In less clear-cut cases, it would be important to express the uncertainty about the results in order to interpret them properly or integrate them into other approaches as you propose in the discussion section.

Minor observations:

- P.1, l.35: typo, sate -> state
- P.2, l.51: typo, casual -> causal
- P.2, l.53 : denies -> denied,
- P.2, l.53 : the 1980s
- P.2, l 57ff: Crucially, counterfactual SCMs in the sense of Pearl and potential outcomes are two sides of the same coin as they can be shown to be logically equivalent, see (Peters et al.)
- P.2, l.69: you should elaborate more clearly here why the models that are mentioned here are not applicable.
- P.3, l.97, typo, spillover7
- P.3, l. 104, proposed -> propose, generalized -> generalize
- P.3, l. 123, typo: bias15
- P.4, l. 148, typo: as well known -> as is well known
- P.5, l. 153, the sentence suddenly ends
- P.5, l. 164, the figure references need to be switched (g)<-> (h)

- P.6, l.186: why does this demonstrate to effectively reveal bidirectional associations? This example is very clearly unidirectional as you also point on a bit further above.
- P.9, l.299: research6
- P.9, l. 309: research45
- P. 9, l. 338 and l 350: casual

Reviewer #2 (Remarks to the Author):

Review of: Inferring causal associations from cross-sectional Earth System data

Gao et al
February 2023

This paper extends convergent cross mapping (CCM) to a spatial dimension, to identify causal associations within “cross-sectional” or spatial datasets. This method is based on constructing manifolds and considering lags as geographical distances between points. The topic is highly relevant in that there are many cases where we have high resolution datasets of spatial information, but limited time-series data points, such that we need methods to draw causal conclusions from these datasets. This was an interesting paper that presented two case studies based on pollution and population to illustrate the utility of the proposed GCCM. However, I found much of the paper and methods difficult to understand, and many details are lacking. With this, I am not convinced that the presented GCCM results make sense, or constitute a robust way to define causal associations from a spatial perspective. These weaknesses are further described below. Unfortunately I cannot recommend this paper for publication at this point in Nature Communications.

Description of GCCM is somewhat lacking, particularly for the meaning and significance of a mapping at a certain “spatial lag”, which is the corollary to a temporal lag in time-series analysis. After reading the paper I still cannot understand why values of rho tend to increase with spatial lags for most relationships in the case studies, since a spatial lag should imply some closeness within an attractor for pixels that are a certain distance away from each other. More generally, the extension of CCM to a spatial context could be interesting, but the lack of details in this paper and weaknesses in the case studies makes it difficult to tell.

This paper would benefit from more definitions that do not rely on the reader being already specialized in CCM.

The case studies based on pollution and population are not very clear, and it is not convincing that the results are novel. Details are missing regarding the actual datasets used (e.g. in the first one, it seems like it is data from China but it is actually from Indiana). Additionally, the conclusions are not particularly interesting, since a causal association is similarly found in all cases in the expected direction, and meanwhile the aspect of associations at different spatial lags are not well explained. These weaknesses are highlighted below in line-by-line comments.

Line by line comments:

Line 28: This is a very long sentence, split up. In general recommend to go through the paper and rephrase several sentences like this.

Line 80: Here is where you first bring up “cross-sectional data” and I am not sure this is the best term, or one that is very familiar. It seems like all your examples are just “spatial data” or something similar to that. I think cross-sectional is specifically referring to different aspects (e.g. population, industry, and pollution) at a snapshot in time, but I think the key aspect in this study is really the

spatial one, hence the "geographic" term in GCCM.

Line 91: "time series presenting" sentence does not make sense

Line 92-101: There are a lot of references to models with no explanations, e.g. Graphical Detector, STUVA, spatial spillover. These made this paragraph difficult to interpret.

Line 104: "resorting to" is an odd choice of wording, would say "used" or similar

Line 123: Here and several places, seems like a referencing typo ("bias15")

Line 126: Here you mention China, which led me to infer your pollution study was based there, however according to the maps in Fig 1 and data statement at the end, it is Indiana and Illinois. This context needs to be more clear.

Line 142: first mention of spatial lags, which are not intuitive. Can a spatial lag be defined as a distance (or radius) away from the point of interest? And what do we expect to see with these relative to the provided result plots? See main comment above on this. Potentially this could use some sort of toy or illustrative example with generated spatial data with a certain dependency embedded.

Line 153: Figure 1 and.

Figure 1: I am not sure how to interpret the x-axis of panels G and H, in that they are the spatial lags. Why is the CCM value increasing with spatial lag for nightlights xmap Cu and industry xmap Cu? Particularly if L can be translated from number of pixels to distances (km), what do these values mean? Similar, if D, E, and F panels are showing the manifolds for just 1 and 2 spatial lags, what should we be reading from these?

Figure 2: These results seem backwards relative to what is described in the paper. Isn't "population density xmap Elevation" the influence of population on elevation? And if so, why is it actually larger than in the opposite direction? These could simply be labeled incorrectly. Additionally, the notation "X xmap Y" is awkward and there could be a better way to write this directional relationship.

I have a similar question as for Figure 1 on how to interpret L on the x-axis, and the increasing "prediction outputs" with L. Finally, in this study, it seems like the different "input" variables behave extremely similarly which makes me suspicious – but I also see that the y-axis are not the same for all panels (panel A has a greater range).

Line 263: The word "complicated ecosystems" is often used, but better to use complex?

Line 364: redundant sentence, states the meaning of "dynamic"

Line 442: "will be illustrated" is strange, since this comes after the main results section.

Methods section: the value "rho" is never defined here, but it is the main vertical axis value of all the results plots.

Line 463: The website with the data (<https://www.resdc.cn>) is all in Chinese, which makes sense, but it seems like for an English-speaking audience there should be a way to obtain the underlying data. As it is, it would be difficult for most readers to reproduce this study. Here, I was trying to see the spatial resolution of the population etc data (details that are missing in the paper) and realized it would not be possible to do that.

Supplementary Information: After reading the paper, I realized some of the information I was looking

for was in the SI, but the SI was not referenced at all the appropriate points. It would be more useful to put the SI into sections and refer to these within the paper, or put critical aspects regarding the datasets into the main paper. The SI also has a very large table that lacks any references.

To Reviewer 1:

Thanks so much for your very professional and detailed comments on our manuscript. Clearly, you are an authority in causation inference, as many comments proved you know the principle, advantage and limitations of mainstream causation models well. Against this background, we feel very encouraged that you gave us positive comment on the innovation, robustness and potential implementations of our new GCCM and we fully believe that your constructive comments can largely improve this manuscript. In the past two months, According to your comment, we have fully revised this manuscript by adding the NPP-climate case and the uncertainty test. Meanwhile, you mentioned that we should consider the influence of random noise in the data sources on the performance of GCCM, which we believe is a very important point. Therefore, we conducted a specific experiment to test the effects of gradually increased noise in the data sources on GCCM outputs. The results proved what you said, the performance of GCCM decreased with the increase of noise. But this experiment also proved GCCM remains valid until the noise was very high. Furthermore, since you suggested the use of such causation models as LiNGaM to compare with GCCM, we also employed LiNGaM for the three cases and LiNGaM failed to extract those causations, which further proved the major advantage of GCCM.

Furthermore, since you suggested some details of this methodology should be better explained, we not only added much details to the revised manuscript, but also largely revised the structure of the manuscript to highlight the major details for an easier and clearer understanding of GCCM.

In general, according to these comments, substantial revisions have been made and the manuscript has been improved significantly. Thank so much again for your constructive comments, which are the essential for making this manuscript qualified for the esteemed NC journal.

Please feel free to contact us if additional revisions are required and we are more willing to conduct any revision accordingly.

Comment 1: In the article under review, the authors adapt convergent cross mapping (CCM), a dynamical systems-based method for inferring cause-effect relationships between time series, to spatial data. As the authors correctly point out in the introduction, although CCM has been applied in several different fields of research (e.g. the Earth sciences), it can often be difficult to obtain time series data of sufficient length, for instance if measurements only started recently compared to the time scale one is interested in. Spatial cross-sectional data on the other hand might be available more abundantly or at least can be obtained more easily. Therefore, the method presented in this work, GCCM, can be a valuable and potentially impactful contribution to the literature on causal inference for dynamical systems.

However, while the idea behind GCCM is original and the real data examples to which the method is applied are interesting, I also have serious criticism concerning the interpretation of the method's result, the discussion of the method's applicability and the presentation of the material.

Regarding the former, as I explain in more detail below, I believe that the results of the first real-data example should be interpreted more carefully. In addition, to be of real value to domain experts, any causality method should be accompanied by a thorough discussion of its underlying assumptions, its limitations, and its failure modes. Unfortunately, such a discussion is mostly lacking here. For example, CCM-based methods have been found to be vulnerable to dynamic noise, see e.g. (Bartsev et al., 2021; Cui, Moore, 2021; Krakovska et al, 2018; Cobey, Baskerville, 2016), and this is likely to be a weakness of GCCM as well. Moreover, the criteria that the method invokes (convergence and $\rho > 0$) are somewhat vague (for instance, when is the method considered to converge exactly?) and it does not come with a measure of uncertainty that conveys to a user how trustworthy the output is. Lastly, the article contains many typos and other minor issues, and the methodology section is not particularly easy to follow.

Response: Thanks so much for your comments. According, we have interpreted the casual relationship in the first real-data case (Line 295-304). For this case, the nightlight was a proxy variable, which reflected residential density and residential waste, well-known as a major cause of soil pollution.

What you mentioned here is very important and we are sorry that we did not make the limitations of GCCM clearer in the previous manuscript. In the revised version, we fully revised the structure and put much detailed introduction of GCCM in the first part of the Results(Line 132-285). In this case, readers can better understand

GCCM. Meanwhile, we have added a specific paragraph to the discussion part concerning the limitations of GCCM (Line 522-542). As revealed by other scholars, the limitations of CCM-based methods proposed mainly include: being sensitive to periodic fluctuations and vulnerable to dynamic noise. The periodic variation is not a main issue for GCCM, as periodic spatial phenomena are very rare in practice. According to your comment, to investigate the influence of noise on GCCM, we added different levels of noise to the heavy metal data (the first case) and ran the GCCM on the synthetic data. By increasing the added random noise from 10% to 30% , 60% and finally 90% of the original observed value, the prediction skill of heavy metals based on industry pollution and residential density decreased. However, GCCM remained robust in identifying the causation with low or medium noise. Only when the added noise reached 60%, GCCM failed to extract causation of residential density on Cu.

The uncertainty test was really a good idea. In the revised manuscript, according to your suggestions, we employed two criteria (the significance and the confidential interval) to demonstrate the uncertainty of the results of GCCM. We have introduced the testing method (Line 251-252) and added it to the corresponding R codes.

In the revised manuscript, according to your comment, we added the definition of convergence(Line 248-251). Since GCCM was the spatial extension of CCM, we had used the original definition of convergence from CCM (Sugihara et al., 2012) as “p increases with the size of libraries and is statistically significant when the library becomes largest”.

We believe your comment that the principle and interpretation of GCCM can be better explained are very important and useful. To address this issue, we have fully re-organized the structure of the manuscript according to similar Method papers in NC, re-written the method section by adding many more details, and invited all authors, including native-speaker scholars to carefully polish the manuscript. Thanks again for all your constructive general and detailed comments.

Comment 2: In conclusion, I unfortunately must vote to reject the publication of this work. Due to the novelty of the idea, a resubmission might be justified if the issues above are addressed properly.

* Bartsev, S., et al. "Imperfection of the convergent cross-mapping method." IOP Conference Series: Materials Science and Engineering. Vol. 1047. No. 1. IOP

Publishing, 2021.

* Cui, Lipeng, and Jack Murdoch Moore. "Causal network reconstruction from nonlinear time series: A comparative study." *International Journal of Modern Physics C* 32.04 (2021): 2150049.

* Krakovska, Anna, et al. "Comparison of six methods for the detection of causality in a bivariate time series." *Physical Review E* 97.4 (2018): 042207.

* Cobey S, Baskerville EB (2016) Limits to Causal Inference with State-Space Reconstruction for Infectious Disease. *PLoS ONE* 11(12): e0169050.

Response: Thanks so much for your encouragement on the innovation of this manuscript. And we have fully revised the manuscript according to all your general and detailed comments and believe it has been largely improved. Thanks again for your great help on our research. These references have all been added to the revised manuscript.

Comment 3: As a major motivating example, you mention the causal link between Net Primary Productivity and precipitation which can't be identified by time series-based causality methods. Yet you don't apply your method to this example and only refer to other spatial methods in the introduction. Is GCCM able to find this causal link?

Response: This is a good idea and thanks so much for pointing it out. In the revised manuscript, we added NPP-climate causation as the third case(Line 422-482), and the result proved GCCM successfully identified the casual links. Meanwhile, LiNGAM, which could be applied to the spatial cross-sectional data, also failed to identify the NPP-climate causation. Thanks again for this constructive comment, which further proved the advantage of GCCM.

Comment 4: At this point, the baseline to which you compare the performance of GCCM is correlation analysis. It would be good to add another baseline method to your analysis that aims to infer causal links. This could be a naïve method, e.g. treating the spatial observations as i.i.d. observations of one random variable and then using a bivariate causality method such as LiNGaM. I expect GCCM to clearly outperform this baseline, but it would be nice to have an empirical demonstration.

Response: Again, this is a good suggestion. In the revised manuscript, we employed LiNGaM as a comparison with GCCM. For the three cases, LiNGaM failed to extract any causations in the first and third case, and simply identified part of causations between population and influencing factors(Line 332-334, 408-412,455-

459). Thanks so much for this valuable comment, which further proved that few, if not none, existing causation models were effective for causation inference in cross-sectional data.

Comment 5: The weaknesses and limitations of GCCM should be discussed more extensively. How vulnerable is the method to noisy data? Did you encounter real world or simulated data examples where GCCM could not identify the correct causal direction? It would be helpful if an example that is a failure mode of the method would be included to illustrate the methods limitations more clearly.

Response: Again, thanks so much for this constructive comment. What you mentioned here is very important and we are sorry that we did not make the limitations of GCCM clearer in the previous manuscript. In the revised version, we fully revised the structure and put many detailed introduction of GCCM in the first part of the Results. In this case, readers can better understand GCCM. Meanwhile, we have added a specific paragraph to the discussion part concerning the limitations of GCCM (Line 522-542). As revealed by other scholars, the limitations of CCM-based methods proposed mainly include: being sensitive to periodic fluctuations and vulnerable to dynamic noise. The periodic variation is not a main issue for GCCM, as periodic spatial phenomena are very rare in practice. According to your comment, to investigate the influence of noise on GCCM, we added different levels of noise to the heavy metal data (the first case) and ran the GCCM on the synthetic data. By increasing the added random noise from 10% to 30% , 60% and finally 90% of the original observed value, the prediction skill of heavy metals based on industry pollution and residential density decreased. However, GCCM remained robust in identifying the causation with low or medium noise. Only when the added noise reached 60%, GCCM failed to extract some causations between heavy metals and industry (residence), which indirectly proved the robustness of GCCM.

Comment 6: The methods section ‘The establishment of GCCM’ feels a bit rushed. It would improve readability if you took more time to explain your notations and disentangle some of the sentences such as in lines 397-400. Computing different quantities explicitly for a toy example would also help.

Response: Thanks so much for this comment. In the revised manuscript, we fully revised the structure of this manuscript and put the description of GCCM earlier in the manuscript (similar to other method papers in NC), so it is easier for readers to understand the methodology of GCCM. In Fig.1 and Fig.2, toy data was

displayed to explain GCCM. Meanwhile, many details have been added to the revised manuscript (Line 132-285), thus we believe the readability of the methodology has been largely improved. Thanks again for your help.

Comment 7: Comments on real data examples: I like the choice of the two real data examples. However, there are additional aspects that should be discussed.

In the first example, the role of proxies should be discussed more clearly. What I mean is that you argue that there is a unidirectional causal link from nightlights to soil pollution whereas clearly, the fact that there is light at night does not pollute the soil. Therefore, in the interventional meaning of causation, the variable nightlights is not a cause of soil pollution. Rather, nightlights are a good proxy for an actual underlying cause, the number of residents in the area. In a way, you don't detect a causal link but a confounded connection between soil pollution and nightlights, the confounder being the number of residents. However, the variable nightlights is so informative about the confounder (i.e. such a close proxy) that it inherits the causal asymmetry between residents and soil pollution. This is a reason why the asymmetry that CCM-based methods such as yours detect should be interpreted more carefully.

Response: Thanks so much for your comment. Yes, your understanding is perfect, and what we meant was exactly the same. We are sorry we did not make this clearer in the previous version. In the revised manuscript, we had made it very clear that nightlight was a proxy variable of the residential density, which was one major influencing factor for heavy metal pollution (Line 295-300). Thanks again for your comment, which helped us address the potential confusion.

Comment 8: In the second example, there seems a mismatch between the labels of the plot and the causal direction that they identify (the labels should be switched). Am I right in assuming that this is not a conceptual issue but rather a mix up when that happened when generating the labels of the causal directions in the plot?

Response: Thank you very much for pointing out this problem. Yes, this is a mixed labeling and we have checked it all through the text and figures, and make sure there was no any potential confusion or mix up in the revised manuscript. Yes, as explained in CCM, $X \mapsto Y$ means the influence of Y on X ($Y \rightarrow X$), which suggests Y is the cause and X is the effect. In the revised manuscript, we have improved the expression of the causation labeling for a better and clear understanding. Thanks so much for your valuable comment, which help us reduce potential confusion.

Comment 9: Is there a way to assess how significant the results of your method are and how much confidence one should attribute to them? I agree that the plots in your examples look convincing but those were relatively clear-cut cases from the start. In less clear-cut cases, it would be important to express the uncertainty about the results in order to interpret them properly or integrate them into other approaches as you propose in the discussion section.

Response: The uncertainty test was really a good idea. In the revised manuscript, according to your suggestions, we employed two criteria (the significance and the confidential interval) to demonstrate the uncertainty of the results of GCCM. We have introduced the testing method (Line 251-252) and added it to the corresponding R codes.

Comment 10:

Minor observations:

- P.1, l.35: typo, sate -> state
- P.2, l.51: typo, casual -> causal
- P.2, l.53 : denies -> denied,
- P.2, l.53 : the 1980s
- P.2. 157ff: Crucially, counterfactual SCMs in the sense of Pearl and potential outcomes are two sides of the same coin as they can be shown to be logically equivalent, see (Peters et al.)

Response: Thanks so much for your comment. We have checked and revised all these typos and checked through the manuscript. We quite agree with you that they were logically equivalent. The main differences between SCMs and potential outputs is the emphasis on casual graph. For the causal graph, Imbens and Rubin (2015) said that “we have not found this approach to aid drawing of causal inferences,”. And Pearl and Mackenzie wrote that “Rubin has steadfastly maintained over the years that diagrams serve no useful purpose”. Therefore, considering that most readers do not have such an incisive understanding of causation models, we divided them into two categories. Again, we agreed with you, and we think that the SCMs and potential outcomes could be combined together, as well as GCCM. We discussed this in the discussion part.

Reference:

Imbens GW, Rubin DB. Causal Inference for Statistics, Social, and Biomedical Sciences: An Introduction. Cambridge University Press (2015).

Pearl J, Mackenzie D. The Book of Why: The New Science of Cause and Effect. Basic

Books (2018).

Comment 11: P.2, l.69: you should elaborate more clearly here why the models that are mentioned here are not applicable.

Response: Thanks so much for your comment. We have revised the statement to better express it(Line 65-67). What we wanted to explain here is that two frameworks are mainly for stochastic processes, while earth system sciences often contain deterministic or dynamical interactions, which breaks the assumption of stochastic processes (e.g. information separability and probability distribution).

Comment 12:

- P.3, l.97, typo, spillover7
- P.3, l. 104, proposed -> propose, generalized -> generalize
- P.3, l. 123, typo: bias15
- P.4, l. 148, typo: as well known -> as is well known
- P.5, l. 153, the sentence suddenly ends
- P.5, l. 164, the figure references need to be switched (g)<-> (h)

Response: Thanks so much for your comment. We have revise the typos and checked through the manuscript.

Comment 13: P.6, l.186: why does this demonstrate to effectively reveal bidirectional associations? This example is very clearly unidirectional as you also point on a bit further above.

Response: Thanks so much for your comment. You are right that it is unidirectional. It was a typo and we have revised it to “unidirectional associations” in the revised manuscript.

Comment 13:

- P.9, l.299: research6
- P.9, l. 309: research45
- P. 9, l. 338 and l 350: casual

Response: Thanks so much for your comment. We have revised the typos and checked through the manuscript.

To Reviewer 2:

Thanks so much for your very professional and detailed comments on our manuscript. And many thanks for the encouragement and acknowledgement from you and another reviewer, who are clearly both experts in spatial analysis and models, on the innovation and robustness of GCCM. Based on a careful analysis of your comments, we found they are all very constructive for use to improve the manuscript. Some your concerns (such as the data availability links) are caused by our unclear expression that easily lead to misunderstanding. In the revised manuscript, we rewrote them to clarify it according to your suggestion.

The most important concern you mentioned is that due to the lack of details, the principle, parameter setting and methodology of GCCM were not clearly explained. Yes, due to limited spaces and a lot of details concerning GCCM setting, which is similar to CCM, we did not explain the setting of GCCM clearly enough. But we fully agreed with you that even if some principles of GCCM have been introduced before in previous paper concerning CCM (Sugihara et al. 2012), it remains necessary for us to explicitly explain our models in this manuscript. Therefore, according to your comment, we not only added many details to the revised manuscript, but also largely revised the structure of the manuscript to highlight the major details for an easier and clearer understanding of GCCM. All your raised issues have been addressed or explained in the revised manuscript.

Meanwhile, according to the suggestions from the other reviewer, we added content concerning another case study, uncertainty test, noise test and another causation model LiNGaM f. Thanks so much to your professional comments. In the past two months, we have made substantial efforts to revise the manuscript accordingly and we believe the manuscript has been improved significantly.

Please feel free to contact us if additional revisions are required and we are more than willing to conduct revisions according to your constructive suggestions.

Comment 1: This paper extends convergent cross mapping (CCM) to a spatial dimension, to identify causal associations within “cross-sectional” or spatial datasets. This method is based on constructing manifolds and considering lags as geographical distances between points. The topic is highly relevant in that there are many cases where we have high resolution datasets of spatial information, but limited time-series data points, such that we need methods to draw causal conclusions from these datasets. This was an interesting paper that presented two case studies based on pollution and

population to illustrate the utility of the proposed GCCM. However, I found much of the paper and methods difficult to understand, and many details are lacking. With this, I am not convinced that the presented GCCM results make sense, or constitute a robust way to define causal associations from a spatial perspective. These weaknesses are further described below. Unfortunately I cannot recommend this paper for publication at this point in Nature Communications.

Description of GCCM is somewhat lacking, particularly for the meaning and significance of a mapping at a certain “spatial lag”, which is the corollary to a temporal lag in time-series analysis. After reading the paper I still cannot understand why values of rho tend to increase with spatial lags for most relationships in the case studies, since a spatial lag should imply some closeness within an attractor for pixels that are a certain distance away from each other. More generally, the extension of CCM to a spatial context could be interesting, but the lack of details in this paper and weaknesses in the case studies makes it difficult to tell. This paper would benefit from more definitions that do not rely on the reader being already specialized in CCM.

Response: Thanks so much for your comment. Yes, due to limited spaces and a lot of similar details to CCM, we did not explain the setting of GCCM clearly enough. But we fully agreed with you that even if some principles of GCCM have been introduced before in previous paper of CCM (Sugihara et al. 2012), it remains necessary for us to explicitly explain our models in this manuscript. Therefore, according to your comment, we not only added many details to the revised manuscript, but also largely revised the structure of the manuscript to highlight the major details for an easier and clearer understanding of GCCM. All your raised issues have been addressed or explained in the revised manuscript.

As you suggested, the spatial lag is indeed the corollary to a temporal lag in time-series. So we have added very detailed description of spatial lag in the revised manuscript and specifically explained it here. In time series data, the lag k means a shift from the observation at focal period t to past observation at $t-k\tau$. Similarly, in spatial data, the lag means a shift to spatial neighbors from the focal spatial unit. The spatial lags are used to compose a vector with the focal point. And the vector are then used to reconstruct the manifold in the state space. As illustrated in Figure 3, due to the weak correlation, the spatial distribution of soil Cu concentrations, the density of industrial pollutants, and the nightlight images present little similarity. But in the reconstructed manifold in the state space, they presented a similar pattern. GCCM is based on the one-to-one map in the reconstructed manifold in the state space. For example, to predict the density of industrial pollutants at spatial unit s , noted as Y_s , GCCM employs the one-to-one mapping

relationship of corresponding points $M_{x,s}$ and $M_{y,s}$ in the reconstructed manifold, which contains s as the first components, and the neighbors of $M_{x,s}$ and $M_{y,s}$. In fact, the lateral axis of the GCCM results is the size of libraries, which means the quantity of observations used in reconstruction of the shadow manifold. We did not clarify this point in previous version, and we introduced this point in detail in this revised version according to your suggestion. For raster data, since spatial units are regularly arranged with equal areas, the window size is used to represent the size of library. For polygon data, the number of spatial units is used to represent the size of library due to the irregular arrangement and unequal areas. With the increase of the libraries, the points in the local area containing $M_{x,s}$ (or $M_{y,s}$) of state space become denser and the cross-mapping prediction becomes more precise, as demonstrated in Fig. 1(c). Therefore, ρ tends to increase with library size when causal associations exist. The details and definitions were added to introduce GCCM more clearly.

Thanks again for your comment, according to which the manuscript has been improved significantly.

Comment 2: The case studies based on pollution and population are not very clear, and it is not convincing that the results are novel. Details are missing regarding the actual datasets used (e.g. in the first one, it seems like it is data from China but it is actually from Indiana). Additionally, the conclusions are not particularly interesting, since a causal association is similarly found in all cases in the expected direction, and meanwhile the aspect of associations at different spatial lags are not well explained. These weaknesses are highlighted below in line-by-line comments.

Response: Thanks so much for your comment.

Firstly, according to your suggestions, we added more details concerning the dataset and deleted the misleading words.

Secondly, we believe there is a major misunderstanding here. Yes, you are right, the extracted metal pollution-industry (residential) causations and population-environment causations were not novel to the society. However, the novel and undiscovered causations were not the focus of this research. Instead, all these cases were employed to test the suitability of GCCM in different situations. As

introduced in our manuscript, it is a common challenge that there is no reliable true data to verify causation models. So the well-accepted prior-knowledge, such as industry-pollution causation, can be used as reference to verify the specific suitability and interpreting capability of GCCM. Therefore, these not-new and well-known causations, as you pointed out, worked effectively to reveal the advantage of GCCM. As the other reviewer, who seems also an authority in causation inference, mentioned he liked the choice of the two cases and he even suggested us to a third case (which is also a well-known causation) to the revised manuscript. According to his/her comments, we added another case concerning farmland NPP-climate (Line 422-482). For the three case studies, GCCM successfully extracted all these well-known causations and effectively identified the direction and strength of these causations based on the cross-sectional data. Meanwhile, these clearly-existing causations cannot be identified by other temporal causation models, such as the commonly employed model LiNGAM (a model suggested by the other reviewer). Generally, since no actual reference data are available for us to properly verify GCCM, we can only consider the strategy to test whether GCCM can identify those clearly existing causations. So, in the revised manuscript, we further emphasized that the aim of these clearly existing, not-new causations was to verify the suitability and robustness of GCCM, which is the major novelty and contribution of this research.

Thanks again for pointing out your concern and we believe that with clearer explanation, readers can better understand our choice of case studies.

Thirdly, according to previous comments, we believe the “different spatial lags” you mentioned here means “different library size”. As explained in details in the above comment, the rho increase with library size when causal associations exist, because with the increase of the libraries, the points in the local area containing $M_{x,s}$ (or $M_{y,s}$) of state space become denser. To avoid potential confusions, we explained this more clearly in the revised manuscript (Line 244-275).

Thanks again for all these comments. By addressing these confusions and misunderstanding, this manuscript has been largely improved.

Comment 3: Line 28: This is a very long sentence, split up. In general recommend to go through the paper and rephrase several sentences like this.

Response: Thanks so much for pointing this out. We have rephrased this sentence

and checked throughout the manuscript to polish the language. Thanks again for this valuable comment.

Comment 4: Line 80: Here is where you first bring up “cross-sectional data” and I am not sure this is the best term, or one that is very familiar. It seems like all your examples are just “spatial data” or something similar to that. I think cross-sectional is specifically referring to different aspects (e.g. population, industry, and pollution) at a snapshot in time, but I think the key aspect in this study is really the spatial one, hence the “geographic” term in GCCM.

Response: Thank you so much for your suggestion. The key aspect in this study is the spatial data. Therefore, we revised the cross-sectional data to spatial cross-sectional data in the revised manuscript.

Comment 5: Line 91: “time series presenting” sentence does not make sense

Response: Thank you so much for your suggestion. We have deleted this sentence in the revised manuscript.

Comment 6: Line 92-101: There are a lot of references to models with no explanations, e.g. Graphical Detector, STUVA, spatial spillover. These made this paragraph difficult to interpret.

Response: Thank you so much for this constructive comment. We have rewritten this part and added explanation to all these key concepts(Line 93-100) .

Comment 7: Line 104: “resorting to” is an odd choice of wording, would say “used” or similar

Response: Thanks for pointing this out. We have revised it accordingly.

Comment 8: Line 123: Here and several places, seems like a referencing typo (“bias15”)

Response: Thanks for pointing this out. We have checked and corrected these.

Comment 9: Line 126: Here you mention China, which led me to infer your pollution study was based there, however according to the maps in Fig 1 and data statement at the end, it is Indiana and Illinois. This context needs to be more clear.

Response: Thanks so much for pointing this out. Yes, the description in the previous version may be misleading and thus we fully revised this paragraph to avoid unnecessary confusions.

Comment 10: Line 142: first mention of spatial lags, which are not intuitive. Can a spatial lag be defined as a distance (or radius) away from the point of interest? And what do we expect to see with these relative to the provided result plots? See main comment above on this. Potentially this could use some sort of toy or illustrative example with generated spatial data with a certain dependency embedded.

Response: Thank you so much for your comment. Despite a limited space, we should still try to make the details of the model setting clear. So as you suggested, many details have been added to the revised manuscript (Line 167-181).

Yes, the spatial lag is defined as a distance (or radius) away from the point of interest. In time series data, the lag k means a shift from the observation at focal period t to past observation at $t-k\tau$. Similarly, in spatial data, the lag means a shift to spatial neighbors from the focal spatial unit. The spatial lags are used to compose a vector with the focal point. To explain the spatial lags, we defined it in details, and illustrated in Fig.1a and b for raster data and polygon data based on some toy data (not the data from three cases). The spatial lags do not join the result plots directly. It is the number of the vectors they composed were used as the lateral axis, i.e. the size of libraries. With the increase of the libraries, the points in the local area containing $M_{x,s}$ (or $M_{y,s}$) of state space become denser and the cross-mapping prediction becomes more precise, as demonstrated in Fig. 1(c). Therefore, ρ tend to increase with library size when causal associations exist.

Thanks so much again for your comment.

Comment 11: Line 153: Figure 1 and.

Response: Thanks for pointing this out. We have checked and corrected these.

Comment 12: Figure 1: I am not sure how to interpret the x-axis of panels G and H, in that they are the spatial lags. Why is the CCM value increasing with spatial lag for nightlights xmap Cu and industry xmap Cu? Particularly if L can be translated from number of pixels to distances (km), what do these values mean? Similar, if D, E, and F panels are showing the manifolds for just 1 and 2 spatial lags, what should we be reading from these?

Response: Thanks for your point. I think there is some unclear parts here and we are giving more details to explain it here and in the revised manuscript (Line 253-

275 and Line 599-629).

The x-axis of panels G and H is the size of libraries, which means the quantity of observations used in reconstruction of the shadow manifold. The spatial lags are used to compose a vector with the focal units. To simplify this problem, here we treat the observation function as the average of spatial lags with the same order. For example $\langle h_s(x), h_{s(1)}(x), \dots, h_{s(L-1)}(x) \rangle$, where the first component is the X value of the focal unit s , $h_{s(1)}(x)$ is the average of X values of first order neighbors of s , $h_{s(2)}(x)$ is the average of X values of second order neighbors of s . The spatial lags are used to form the elements of the vectors, i.e. the dimension of points in Fig.3 d-f (The Fig.1 in previous version), while the size of library means to the number of points in Fig.3 d-f. Fig.3 d-f plotted all observations, i.e. the largest size of library. From Fig.3 d-f, it could be found that, due to the weak correlation, the spatial distribution of soil Cu concentrations, the density of industrial pollutants, and the nightlight images present little similarity. But in the reconstructed manifold in the state space, they presented a similar pattern. Therefore we can predict them based on the one-to-one map in the reconstructed manifold in the state space. For example, to predict the density of industrial pollutants at spatial unit s , noted as Y_s , GCCM employs the one-to-one mapping relationship of corresponding points $M_{x,s}$ and $M_{y,s}$ in the reconstructed manifold, which contains s as the first components, and the neighbors of $M_{x,s}$ and $M_{y,s}$. With the increase of the libraries, the points in the local area containing $M_{x,s}$ (or $M_{y,s}$) of state space become denser and the cross-mapping prediction becomes more precise, as demonstrated in Fig. 1(c). Therefore, ρ tends to increase with library size when causal associations exist.

Thanks again for your comment. The details and definitions were added to introduce GCCM more clearly (Line 132-285 and 599-629).

Comment 13: Figure 2: These results seem backwards relative to what is described in the paper. Isn't "population density xmap Elevation" the influence of population on elevation? And if so, why is it actually larger than in the opposite direction? These could simply be labeled incorrectly. Additionally, the notation "X xmap Y" is awkward and there could be a better way to write this directional relationship. I have a similar

question as for Figure 1 on how to interpret L on the x-axis, and the increasing “prediction outputs” with L. Finally, in this study, it seems like the different “input” variables behave extremely similarly which makes me suspicious – but I also see that the y-axis are not the same for all panels (panel A has a greater range).

Response: Yes, thanks so much for your careful checking and we acknowledged this was a typo (or a mixed labelling). In the revised manuscript, we have checked carefully and make sure no such mixed use happened again. Meanwhile, to make the expression clearer and avoid potential confusions, we further improved the expression in the revised manuscript according to your good advice.

As defined in CCM, $X \text{ xmap } Y$ is a specific defined term (Sugihara et al, 2012), means the influence of Y on X ($Y \rightarrow X$), which indicates that Y is the cause and X is the effect.

So Yes, “Population density xmap Elevation” means the casual influence of Elevation on population density (Elevation \rightarrow population density).

By revising the expression of causations, especially causation direction in the revised manuscript, readers can better understand the direction of causations between variables.

Thanks so much again for your constructive comment.

Comment 14: Line 263: The word “complicated ecosystems” is often used, but better to use complex?

Response: Thank you so much for your suggestion. We have corrected accordingly.

Comment 15: Line 364: redundant sentence, states the meaning of “dynamic”

Response: Thank you so much for your suggestion. We have corrected accordingly.

Comment 16: Line 442: “will be illustrated” is strange, since this comes after the main results section.

Response: Thank you so much for your suggestion. We have corrected accordingly.

Comment 16: Methods section: the value “rho” is never defined here, but it is the main vertical axis value of all the results plots.

Response: Thank you so much for your suggestion. We have corrected accordingly. We defined the “rho” in the newly added Equation (7) and explained it in the

revised manuscript.

Comment 17: Line 463: The website with the data (<https://www.resdc.cn>) is all in Chinese, which makes sense, but it seems like for an English-speaking audience there should be a way to obtain the underlying data. As it is, it would be difficult for most readers to reproduce this study. Here, I was trying to see the spatial resolution of the population etc data (details that are missing in the paper) and realized it would not be possible to do that.

Response: Thanks for pointing it out. I think there is a major misunderstanding here. In the previous manuscript, we had already uploaded a specific dataset, which included all the data sources used for this research and English data description, and introduced it in the data availability part as “The Code for GCM and the data employed for this research are available at <https://doi.org/10.6084/m9.figshare.21782201>”. The website with the data (<https://www.resdc.cn>) is the original source we got the climate data and we listed it just to demonstrate its authority. The default language for this website is Chinese, which may cause such difficulties. To avoid such misunderstanding, we listed the website with English instead (Line 586-591).

So generally, all readers can easily get the required data from the dataset we uploaded or directly download it from the website.

Thanks again for raise your concern.

Comment 18: Supplementary Information: After reading the paper, I realized some of the information I was looking for was in the SI, but the SI was not referenced at all the appropriate points. It would be more useful to put the SI into sections and refer to these within the paper, or put critical aspects regarding the datasets into the main paper. The SI also has a very large table that lacks any references

Response: Thanks so much for your comment. According to your comment, for a clearer understanding, we not only moved many contents to the main text from SI, but also largely restructured the manuscript and added many details. We also added many references to SI tables as you suggested.

Thanks again for advising this, which improved the readability of this paper significantly.

REVIEWER COMMENTS

Reviewer #1 (Remarks to the Author):

Answers to rebuttal/ comments on first revised version:

In the revised version of their submission, the authors have incorporated several changes that significantly improve their work compared to the original version. Nevertheless, some issues remain, and these should be addressed before I can recommend this paper for publication. Given that the authors have been able to improve the previous version in several places, I would encourage them to provide a second revised version that addresses the remaining problems.

Below, when referring to earlier comments, I will employ the numbering that you used in your rebuttal to the first review.

Major comments to the authors:

- Compared to the earlier draft, language and grammar have clearly improved in the main document. There are still several grammatical mistakes and imprecise formulations, some of which I point out below, but these affect readability much less than before. Unfortunately, the supplement (especially the first section) is still in a poor state language-wise and this does seriously affect intelligibility. This section needs to be thoroughly edited for language structure and clarity, or with the help of software tools. In addition, the section feels rushed, and you use a lot of terminology without explaining it. I would suggest that you either explain different concepts and assumptions in more detail, or you only mention their existence very shortly and refer to existing work for detailed explanations. Currently, you use an in-between approach which is often hard to follow.
- Comments 3,4 & 5: I like that you included the NPP – precipitation example and that you evaluated the soil pollution example once more with different levels of added noise. Similarly, adding LinGaM as an additional baseline strengthens your argument. These changes improve the paper substantially in my opinion.
- Comment 6: The section on methodology that explains the theory behind GCCM has improved and it was a good idea to move it to an earlier section. However, it is still hard to digest and sometimes too hand-wavy on the mathematics behind the method. I think you should explain the mathematical details in much larger detail and give more precise formal definitions of the objects you employ. It would also help if some of the underlying results, such as the generalization of Takens' theorem were included, at least in the supplement. From a mathematician's point of view, some choices of notations are also suboptimal. This is the most fundamental section of your article, and it warrants a more careful and precise discussion.
- Comment 7: I think there is a bit of a misunderstanding here. The fact that your method finds a causal link nightlights \rightarrow soil pollution is a challenge to the interpretation of your method's output. Nightlights are not a cause of soil pollution in the interventional sense of Pearl: If we would intervene on nightlights by switching them off, this would not reduce soil pollution by itself. Rather the correct causal graph is something like nightlights \leftarrow residential density \rightarrow soil pollution and the asymmetry you detect is a consequence of the fact that the association between nightlights and residential density is very strong. If you applied your method on new data (X,Y) without much background knowledge and you find a link $X \rightarrow Y$, you cannot claim that this is a causal link, but only that X is sufficiently close to a cause X' of Y. This is still useful information and does not invalidate your method, but it needs to be discussed much more thoroughly than what you currently do.
- Comment 8: fixed.
- Comment 9: I am glad that you added a significance test to your method. However, this warrants more than two lines of text. The explanation on how you test is too hand-wavy in my opinion as this is crucial information. Please be more specific on the following: what do you mean by ' ρ is statistically significant when the library becomes largest'? Do you mean ' ρ is significantly different from 0'? What is your test statistic for this and what is your choice of significance level? How do you measure

concretely whether there is an increase of ρ with increasing size of L ?

- Comment 10 – 13: fixed.

Minor comments:

- For some reason, the abstract is written in past tense. I would suggest switching to present tense. There are also some sentences in the introduction where you should change to present tense, e.g. line 151.
- Abstract: Causation inference in complicated systems -> Causal inference in complex systems. Generally, the common term in the field is causal inference, not causation inference.
- Abstract: split last sentence in two.
- Line 44: the prerequisite -> a prerequisite
- l. 58: 'the strength of the cause' -> the strength of the causal relationship
- l. 60: 'the randomized control trial' -> randomized control trials
- l. 67: '... which render these two frameworks unapplicable'. I find this too harsh, I would rather say '... which are not captured well by the assumptions of these frameworks and therefore challenge their applicability.'
- l. 69: 'the spatial heterogeneity' -> lose 'the'
- l. 107: again, lose 'the' before the theories
- l. 133: this sentence is strange, what do you mean by 'internal essences'?
- l.158: consists -> is
- l.161: dynamic system -> dynamical system; different from -> in contrast to
- l.178: lags of current orders is strange terminology, maybe explain this sentence more clearly.
- L.224: missing bracket)
- P.6: I still find your notation conventions quite confusing, especially the notation $M_{\{x,s_i\}}$. You explain a lot in words, but I would like to see cleaner mathematical definitions of the objects that you use. Why don't you work more with the embedding map Ψ_h for instance? As far as I understand, the only thing you do here is to define a metric d on the manifold M_x by $d(w,z) = d'(\Psi_h(w), \Psi_h(z))$ for arbitrary points w,z on M_x , where d' is an averaged sum $1/L \sum d_s$. over the Euclidean metrics d_s on $R^{\{\text{lag } s \text{ neighbors}\}}$. This is well-defined as Ψ_h is an embedding.
- Equation (6): with this definition of h , I don't see why $\|\dots\|$ defines a norm in a mathematical sense, and consequently why $\text{dis}()$ defines a metric on M_x . More precisely, can you explain why $\|\dots\|$ or $\text{dis}()$ satisfy the property of positivity of metrics here? Or is being a metric not essential? In the latter case, use a different notation than $\|\dots\|$ as this strongly suggests that this is a norm in a mathematical sense. Again, it would help to have clearer notation in this section.
- L.254: 'it is a required condition'. Required for what exactly?
- L. 448: in-slaved, do you mean enslaved?
- L. 526: 'the periodic time series' -> 'periodic time series'

Reviewer #2 (Remarks to the Author):

Second review of: Inferring causal associations from cross-sectional Earth System Data

I think the authors did a very nice job making improvements to this paper, and largely addressed comments and suggestions from my own and another review. The paper reads a lot more clearly as the method is more understandable, and the case studies are better presented as "tests" rather than novel research findings using the GCCM method. There is also a clearer motivation for spatial causality techniques like this. With that, I think this paper would be acceptable for publication after some minor/moderate revisions noted below. Mainly I feel like the figures could be improved – some figures could be combined into more complex figures, and figures could also show the comparison of GCCM results with the more traditional methods.

General comment: Throughout the paper, I found that past and present tense were combined, and would recommend to use present tense for everything the authors did (e.g. We found, proposed, discovered, this was, these were  present tense). However this is mainly stylistic.

Line 89: Here a study on NPP is alluded to but not referenced (I think it gets referenced later in the paper)

Line 122: Should this little paragraph be labeled "Results"? Maybe this is a journal specific section that requires a short review of the results that are explained later.

Equation 4: missing a parenthesis

Line 266: Here you mention that $X \mapsto Y$ is the basis to determine whether Y causes X (so $Y \rightarrow X$, or " Y causes X "). To me this is counter-intuitive, but useful to state explicitly like this. Later (page 18) this aspect is repeated and it is noted that it may easily be mistaken – it might be helpful to re-iterate at different points in the paper such as in a figure caption (e.g. "Note that the higher $Cu \mapsto$ Industry indicates a causal effect from industry to soil pollution")

Figure 2: It seems like panel c in Figure 1 would go well with this figure – maybe Figure 1c and Fig 2a belong together rather than separate figures?

Figure 3: in panel (g), should Cd be Cu?

Line 311: demonstrate

Line 368: Here is a case where the value does not fit into the confidence interval? " $0.48(p=0.00; 95\% \text{ confidence interval} = [0.61, 0.66])$ ".

In general, I think the p-values are useful, but the confidence intervals make the sentences difficult to read and don't contribute to understanding the results. I would shift to presenting these results in table form if you want to include all p-values and CI for each relationship. Alternately, it would be most useful to have those values, and the values of linear correlation and LinGam metrics listed directly inside the white space in each figure panel. This would help illustrate the differences between the metrics and whether the baseline models do or do not find a correlation for each case so the reader can refer to the figures and not have to sift through paragraphs for the statistical significance and comparison.

Line 392: "Particulate Matters" should probably be "particulate matter"

Figure 4: formatting issue with vertical line around panel (d). Also, it seems like this figure could be combined into one potentially more informative figure, with 8 lines (e.g. 4 colors for the different variables, and 2 line styles, or something similar). Finally, suggest to change "slop" to "slope" in figure title.

Figure 4 discussion: Explanations are discussed for different possible mechanisms behind the finding of significant but smaller $Y \mapsto$ Population (where $Y =$ precip, temp, elev, or slope), but it should be noted that GCM doesn't actually distinguish between any of these possible reasons (for example the fact that population could have a physical feedback on air temperature, but not so much with elevation).

Reference 53: some typo in this reference

Reviewer #1 (Remarks to the Author):

- Compared to the earlier draft, language and grammar have clearly improved in the main document. There are still several grammatical mistakes and imprecise formulations, some of which I point out below, but these affect readability much less than before. Unfortunately, the supplement (especially the first section) is still in a poor state language-wise and this does seriously affect intelligibility. This section needs to be thoroughly edited for language structure and clarity, or with the help of software tools. In addition, the section feels rushed, and you use a lot of terminology without explaining it. I would suggest that you either explain different concepts and assumptions in more detail, or you only mention their existence very shortly and refer to existing work for detailed explanations. Currently, you use an in-between approach which is often hard to follow.

R: Thanks so much for your kind encouragement. We found your comments during the first and second-round review are most professional and constructive. By carefully revising the manuscript fully according to all your detailed comments, our manuscript has been largely improved. We are very glad to have you, clearly a top mathematician who are familiar with causation models and CCM, as the most suitable reviewer for this manuscript.

Specifically, thanks so much for suggesting us for further improving the supplemental materials. Yes, we realized that some terminology should be clearly explained to let readers have a better understanding of the background of casual inference methods. So according to your comments, we have added many details to the supplemental materials and carefully polished the English. Therefore, the revised SI has been largely enhanced.

Thanks again for helping us improving this manuscript. Please feel free to contact us if additional revisions are required.

- Comments 3,4 & 5: I like that you included the NPP – precipitation example and that you evaluated the soil pollution example once more with different levels of added noise. Similarly, adding LinGaM as an additional baseline strengthens your argument. These

changes improve the paper substantially in my opinion.

R: Thanks so much for your encouragement. Yes, your suggestion on adding the NPP case and the LiNGam were great ideas. They improved readers' understanding of the wide suitability and advantage of GCCM and largely enhanced the original manuscript. Thanks again for all these constructive comments.

- Comment 6: The section on methodology that explains the theory behind GCCM has improved and it was a good idea to move it to an earlier section. However, it is still hard to digest and sometimes too hand-wavy on the mathematics behind the method. I think you should explain the mathematical details in much larger detail and give more precise formal definitions of the objects you employ. It would also help if some of the underlying results, such as the generalization of Takens' theorem were included, at least in the supplement. From a mathematician's point of view, some choices of notations are also suboptimal. This is the most fundamental section of your article, and it warrants a more careful and precise discussion.

R: Thanks so much for this comment. We have followed your suggestions and double-checked all the notations in the manuscript. We found that you were indeed an excellent mathematician and your comments are precise and very valuable. Again, we are very grateful to have you as the reviewer, who helped us improve this manuscript significantly. We have rewritten the methodology section by adding more details of the method. We have carefully read the original works explaining Takens' theorem with time series and the generalized embedding theorem to present a concise, yet precise and informative introduction of the two theory in the methodology part. In addition, we have checked all relevant references and realized that you are quite right. We have carefully revised all notations according to your comments and believe these equations are largely improved.

Thanks again for these wonderful and professional comments.

- Comment 7: I think there is a bit of a misunderstanding here. The fact that your method

finds a causal link nightlights → soil pollution is a challenge to the interpretation of your method's output. Nightlights are not a cause of soil pollution in the interventional sense of Pearl: If we would intervene on nightlights by switching them off, this would not reduce soil pollution by itself. Rather the correct causal graph is something like nightlights ← residential density → soil pollution and the asymmetry you detect is a consequence of the fact that the association between nightlights and residential density is very strong. If you applied your method on new data (X, Y) without much background knowledge and you find a link X → Y, you cannot claim that this is a causal link, but only that X is sufficiently close to a cause X' of Y. This is still useful information and does not invalidate your method, but it needs to be discussed much more thoroughly than what you currently do.

R: Thanks so much for pointing this out. This is a very good point. We attempted to establish the casual links between residence and soil pollution. But due to the lack of residential density data, we chose the strategy of employing nightlight intensity data as the proxy variable. Actually, nightlight has been widely accepted as an effective reflection and employed as a proxy variable of residence density (Wang et al. ,2018; Tan et al.,2018; etc). Yes, you are right, the reason and validity of using the nightlight as the proxy value should be clearly explained. So in the revised manuscript, we have added relevant references and revised the expression to avoid unnecessary misunderstanding. Meanwhile, we have referred nightlight as residence throughout the manuscript to better explain the causation.

Thanks again for this constructive comment, which made the methodology part much clearer.

Wang L, et al. Mapping population density in China between 1990 and 2010 using remote sensing. *Remote Sensing of Environment* 210, 269-281 (2018).

Tan M, et al. Modeling population density based on nighttime light images and land use data in China. *Applied Geography* 90, 239-247 (2018).

- Comment 8: fixed.
- Comment 9: I am glad that you added a significance test to your method. However, this warrants more than two lines of text. The explanation on how you test is too hand-wavy in my opinion as this is crucial information. Please be more specific on the following: what do you mean by 'rho is statistically significant when the library

becomes largest'? Do you mean 'rho is significantly different from 0'? What is your test statistic for this and what is your choice of significance level? How do you measure concretely whether there is an increase of rho with increasing size of L?

R: Thanks so much for this comment. We added the details of the significance test and confidence interval estimation. The statistical significance means that rho is significantly different from zero when the library becomes largest. The significance level in the case studies is set to 0.05 and the confidence interval is 95%. Following CCM, the increase of rho with increasing size of L is determined by plotting a line graph of rho vs L.

- Comment 10 – 13: fixed.

Minor comments:

- For some reason, the abstract is written in past tense. I would suggest switching to present tense. There are also some sentences in the introduction where you should change to present tense, e.g. line 151.

R: Thanks so much for this comment. Corrected.

- Abstract: Causation inference in complicated systems -> Causal inference in complex systems. Generally, the common term in the field is causal inference, not causation inference.

R: Thanks so much for this comment. Revised, and we replaced all "Causation inference" with "causal inference"

- Abstract: split last sentence in two.

R: Thanks so much for this comment. Corrected.

- Line 44: the prerequisite -> a prerequisite

R: Thanks so much for this comment. Corrected.

- l. 58: 'the strength of the cause' -> the strength of the causal relationship

R: Thanks so much for this comment. Corrected.

- l. 60: 'the randomized control trial' -> randomized control trials

R: Thanks so much for this comment. Corrected.

- l. 67: '... which render these two frameworks unapplicable'. I find this too harsh, I would rather say '... which are not captured well by the assumptions of these frameworks and therefore challenge their applicability.'

R: Thanks so much for this comment and advising us how to improve it. We have corrected it accordingly.

- l. 69: 'the spatial heterogeneity' -> lose 'the'

R: Thanks so much for this comment. Corrected.

- l. 107: again, lose 'the' before the theories

R: Thanks so much for this comment. Corrected.

- l. 133: this sentence is strange, what do you mean by 'internal essences'?

R: Thanks so much for pointing this out. We have corrected according to your suggestion.

- l.158: consists -> is

R: Thanks so much for this comment. Corrected.

- l.161: dynamic system -> dynamical system; different from -> in contrast to

R: Thanks so much for this comment. Corrected.

• L.178: lags of current orders is strange terminology, maybe explain this sentence more clearly.

R: Thanks so much for this comment. We have revised it into “the first-order lags are adjacent units sharing common edges or vertexes with the focal unit, and the lags of next order are the first-order lags of those adjacent units (excluding those already included)”.

• L.224: missing bracket)

R: Thanks so much for this comment. Corrected.

• P.6: I still find your notation conventions quite confusing, especially the notation $M_{\{x,si\}}$. You explain a lot in words, but I would like to see cleaner mathematical definitions of the objects that you use. Why don't you work more with the embedding map Ψ_h for instance? As far as I understand, the only thing you do here is to define a metric d on the manifold M_x by $d(w,z) = d'(\Psi_h(w), \Psi_h(z))$ for arbitrary points w, z on M_x , where d' is an averaged sum $1/L \sum d_s$. over the Euclidean metrics d_s on $R^{\{|\text{lag } s \text{ neighbors}| \}}$. This is well-defined as Ψ_h is an embedding.

R: Thanks so much for this comment. As explained above, we are very grateful that you gave us these very professional and helpful comments. We have re-checked all required documents and revised all related notations according to your suggestions. Again, thanks so much for this great comments.

• Equation (6): with this definition of h , I don't see why $\|\dots\|$ defines a norm in a mathematical sense, and consequently why $\text{dis}()$ defines a metric on M_x . More precisely, can you explain why $\|\dots\|$ or $\text{dis}()$ satisfy the property of positivity of metrics here? Or is being a metric not essential? In the latter case, use a different notation than $\|\dots\|$ as this strongly suggests that this is a norm in a mathematical sense. Again, it would help to have clearer notation in this section.

R: Thanks so much for this comment. Again, we are very impressed and grateful for giving us this constructive comments. Yes, we checked this and realized that you are quite right. Accordingly, in the revised manuscript, we replaced $\|\dots\|$ into

an function called abs. Thank you very much for your reminder.

- L.254: 'it is a required condition'. Required for what exactly?

R: Thanks so much for this comment. It is required for that the causal associations between X and Y are bidirectional or they are both the effects of a shared cause. That it is a necessary condition for the fact after the “when” (the causal associations between X and Y are bidirectional or they are both the effects of a shared cause). We have rephrased this sentence.

- L. 448: in-slaved, do you mean enslaved?

R: Thanks so much for this comment. Yes, it should be enslaved.

- L. 526: 'the periodic time series' -> 'periodic time series'

R: Thanks so much for this comment. Corrected.

To Reviewer #2 (Remarks to the Author):

R: Thanks so much for acknowledging our efforts on revising this manuscript and your very kind encouragement. We found your comments during the first and second-round review are most professional and constructive. By carefully revising the manuscript fully according to all your detailed comments, our manuscript has been largely improved. We are very glad to have you as a very suitable reviewer for this manuscript. Thanks again for helping us improving this manuscript. Please feel free to contact us if additional revisions are required.

General comment: Throughout the paper, I found that past and present tense were combined, and would recommend to use present tense for everything the authors did (e.g. We found, proposed, discovered, this was, these were  present tense). However this is mainly stylistic.

R: Thanks so much for this comment. This is a very good suggestion. We have checked throughout the whole paper and revised the tense fully according to your comment. Thanks again for pointing this out, which improved the manuscript a lot.

Line 89: Here a study on NPP is alluded to but not referenced (I think it gets referenced later in the paper)

R: Thanks so much for this comment. We have added the reference to the study on NPP.

Line 122: Should this little paragraph be labeled “Results”? Maybe this is a journal specific section that requires a short review of the results that are explained later.

R: Thanks so much for this comment. We labeled “Results” by referring to published papers on Nature Communications. Thanks again for pointing this out.

Equation 4: missing a parenthesis

R: Thanks so much for this comment. We have corrected it accordingly.

Line 266: Here you mention that $X \mapsto Y$ is the basis to determine whether Y causes X (so $Y \rightarrow X$, or “ Y causes X ”). To me this is counter-intuitive, but useful to state

explicitly like this. Later (page 18) this aspect is repeated and it is noted that it may easily be mistaken – it might be helpful to re-iterate at different points in the paper such as in a figure caption (e.g. “Note that the higher Cu xmap Industry indicates a causal effect from industry to soil pollution”)

R: Thanks so much for this comment. Yes, as you suggested in this-round and first-round, this part may cause confusion to readers. Therefore, according to your comment, we tried to repeat this point at different parts throughout the revised manuscript to remind readers the counter-intuitive results. Thanks again for this good advice.

Figure 2: It seems like panel c in Figure 1 would go well with this figure – maybe Figure 1c and Fig 2a belong together rather than separate figures?

R: Thanks so much for this comment. We removed Figure 1c in to Fig 2, and put them together to illustrate the mutual neighbors of bidirectional and unidirectional associations. Thanks again for this good advice.

Figure 3: in panel (g), should Cd be Cu?

R: Thanks again for pointing it out. Yes, It is Cu, we have corrected it.

Line 311: demonstrate

R: Thanks so much for this comment. Corrected.

Line 368: Here is a case where the value does not fit into the confidence interval? “0.48(p=0.00; 95% confidence interval= [0.61,0.66])”.

R: Thanks so much for this comment. We have corrected it accordingly.

In general, I think the p-values are useful, but the confidence intervals make the sentences difficult to read and don't contribute to understanding the results. I would shift to presenting these results in table form if you want to include all p-values and CI for each relationship. Alternately, it would be most useful to have those values, and the values of linear correlation and LinGam metrics listed directly inside the white space in each figure panel. This would help illustrate the differences between the metrics and whether the baseline models do or do not find a correlation for each case so the reader can refer to the figures and not have to sift through paragraphs for the statistical significance and comparison.

R: Thanks so much for your good comment. It really helps. We have deleted all confidence intervals with in the sentence in the main text, while readers can also find them from the supplementary tables. And your suggestions on adding result of correlation and LinGAM to the figure is very useful, as it can give readers a very clear picture of the comparison between GCCM, correlation analysis and LinGAM. Since correlation and LinGAM only provided one single value, instead of a figure, which may not be properly added to the GCCM figures with a good visual effects. And we cannot directly list the value of correlation and LinGAM on the figure of GCCM, otherwise it may cause confusion for readers when reading GCCM outputs. In this case, we added the value of correlation and LinGAM to the detailed note in figures. Consequently, readers can immediately get the comparison between GCCM, correlation analysis and LinGAM without checking Supporting information. Thanks again for this valuable comments.

Line 392: “Particulate Matters” should probably be “particulate matter”

R: Thanks so much for this comment. Corrected.

Figure 4: formatting issue with vertical line around panel (d). Also, it seems like this figure could be combined into one potentially more informative figure, with 8 lines (e.g. 4 colors for the different variables, and 2 line styles, or something similar). Finally, suggest to change “slop” to “slope” in figure title.

R: Thanks so much for this comment. Yes, your suggestions can help us achieve a better explanation effect. Accordingly, we have revised to plot and the title. We combined four plots into one, and foud it is not easy to distinguish them. As in the whole paper, we keep the same style that the orange color is the main casual direction, and the blue is not, we retained the four plots to make fast interpretation feasible. Thanks again for you good advice.

Figure 4 discussion: Explanations are discussed for different possible mechanisms behind the finding of significant but smaller Y xmap Population (where Y = precip, temp, elev, or slope), but it should be noted that GCCM doesn't actually distinguish between any of these possible reasons (for example the fact that population could have a physical feedback on air temperature, but not so much with elevation).

R: Thanks so much for this comment. In the strong coupling cases, GCCM itself

cannot distinguish the feedback causation or reflection. GCCM can only identify the leading direction of the casual associations. To decide whether the weaker coupling was a feedback causation or a reflection caused by the strong enslaved effects, sufficient prior-knowledge or further investigation are required. Yes, this is a very good point when employing GCCM. Accordingly, we have specifically added this to the discussion part to give instructions to readers.

Thanks again for this constructive comments.

Reference 53: some typo in this reference

R: Thanks so much for this comment. We have fixed typos in this reference.

REVIEWERS' COMMENTS

Reviewer #1 (Remarks to the Author):

Thank you for incorporating my suggestions into your work. Overall, I believe that the manuscript has improved a lot since its original submission and may be suited for publication in Nature Communications.

I will address the three major open issues that I mentioned in my previous comments:

Mathematical notation: I am still not fully convinced by some of your notational choices, but I think that the formal description of your methodology has improved enough to be acceptable. As a minor suggestion, maybe give your $\text{abs}()$ functions in equations (5) and (6) subscripts to make clear that these are distinct from each other.

The role of proxies: I am still under the impression that you are not fully understanding the exact nature of my criticism here. I am *not* questioning that nightlights are a good proxy for residential density. Instead, I am making the point that the use of proxies is a challenge to interpreting the asymmetry that your method finds as *causation*. At least that is the case, if we interpret causation as intervening on X induces a change in Y. In fact, we know for sure that intervening on the proxy in your example (X = nightlights) does not induce a change in Y (= pollution). In this particular example, it might be clear what the actual underlying cause is (many residents) but in other examples this might be much less clear. And these other examples may be exactly what people would like to use your method for! After all, we want to use it in cases where we don't know the causal relationship a priori already, as in your toy example!

I do not expect you to write a philosophical text on the role of proxies in causality but at least it would have been nice if this issue was acknowledged. However, if you prefer not to do this, *I will accept that* since this point is not significant enough to keep your work from being published. I would ask you though to reverse the change from nightlights to residence as this change is simply hiding the proxy issue, rather than addressing it.

Supplement: The first section of the supplement has clearly improved, good work on that!

Reviewer #1 (Remarks to the Author):

Thank you for incorporating my suggestions into your work. Overall, I believe that the manuscript has improved a lot since its original submission and may be suited for publication in Nature Communications.

R: Thanks so much for your encouragement and great help during three-round review, without which the quality of this manuscript cannot meet the high-standard of NC.

I will address the three major open issues that I mentioned in my previous comments:

Mathematical notation: I am still not fully convinced by some of your notational choices, but I think that the formal description of your methodology has improved enough to be acceptable. As a minor suggestion, maybe give your $\text{abs}()$ functions in equations (5) and (6) subscripts to make clear that these are distinct from each other.

R: We have revised the notations in equations (5) and (6) according to your suggestions. Thanks so much for your comment in this-round and last-round review, which helped improve this manuscript so much. .

The role of proxies: I am still under the impression that you are not fully understanding the exact nature of my criticism here. I am *not* questioning that nightlights are a good proxy for residential density. Instead, I am making the point that the use of proxies is a challenge to interpreting the asymmetry that your method finds as *causation*. At least that is the case, if we interpret causation as intervening on X induces a change in Y. In fact, we know for sure that intervening on the proxy in your example (X = nightlights) does not induce a change in Y (= pollution). In this particular example, it might be clear what the actual underlying cause is (many residents) but in other examples this might be much less clear. And these other examples may be exactly what people would like to use your method for! After all, we want to use it in cases where we don't know the causal relationship a priori already, as in your toy example!

I do not expect you to write a philosophical text on the role of proxies in causality but at least it would have been nice if this issue was acknowledged. However, if you prefer not to do this, *I will accept that* since this point is not significant enough to keep your work from being published. I would ask you though to reverse the change from nightlights to residence as this change is simply hiding the proxy issue, rather than addressing it.

R: We are sorry that we did not fully get you in the last-round revisions. We have reverse the use of residence to nightlight in the revised manuscript according to your comment. Meanwhile, in the revised manuscript (Discussion part), we have admittedly that we used a proxy variable in the first case to reveal an already known causation, which should be carefully conducted for other scholars. We also suggested that there are some criteria to follow for properly selecting agent variables. Thanks so much again for your clarification.

Supplement: The first section of the supplement has clearly improved, good work on that!

R: Thanks so much for your encouragement.